# Myostatin regulates fatty acid desaturation and fat deposition through MEF2C/miR222/SCD5 cascade in pigs

Hongyan Ren[1,5], Wei Xiao[1,5], Xingliang Qin[2], Gangzhi Cai[1], Hao Chen[1], Zaidong Hua[1], Cheng Cheng[2], Xinglei Li[3], Wenjun Hua[1], Hongwei Xiao[1], Liping Zhang[1], Jiali Dai[2], Xinmin Zheng[1], Zhe Zhu[1], Chong Qian[4], Jie Yao[2✉] & Yanzhen Bi [1✉]

Myostatin (MSTN), associated with the "double muscling" phenotype, affects muscle growth and fat deposition in animals, whereas how MSTN affects adipogenesis remains to be discovered. Here we show that MSTN can act through the MEF2C/miR222/SCD5 cascade to regulate fatty acid metabolism. We generated MSTN-knockout (KO) cloned Meishan pigs, which exhibits typical double muscling trait. We then sequenced transcriptome of subcutaneous fat tissues of wild-type (WT) and MSTN-KO pigs, and intersected the differentially expressed mRNAs and miRNAs to predict that stearoyl-CoA desaturase 5 (SCD5) is targeted by miR222. Transcription factor binding prediction showed that myogenic transcription factor 2C (MEF2C) potentially binds to the miR222 promoter. We hypothesized that MSTN-KO upregulates MEF2C and consequently increases the miR222 expression, which in turn targets SCD5 to suppress its translation. Biochemical, molecular and cellular experiments verified the existence of the cascade. This novel molecular pathway sheds light on new targets for genetic improvements in pigs.

[1] Key Laboratory of Animal Embryo Engineering and Molecular Breeding of Hubei Province, Institute of Animal Science and Veterinary Medicine, Hubei Academy of Agricultural Sciences, 430064 Wuhan, China. [2] Wuhan Biojie Biomedical and Technology Co., Ltd., 430000 Wuhan, China. [3] Wuhan Bioacme Biotechnology Co., Ltd., 430000 Wuhan, China. [4] Beijing Center for Physical and Chemical Analysis, 100094 Beijing, China. [5]These authors contributed equally: Hongyan Ren, Wei Xiao. ✉email: yaojie@biojust.cn; sukerbyz@126.com

MSTN, also known as growth/differentiation-8, is an important member of the transforming growth factor β family. Functional blockade of MSTN results in the double muscling trait in animals, which features overgrown skeletal muscle and reduced fat mass. Owning to this advantage, MSTN has been a prime genetic target for the development of high-leanness animal cultivars in livestock, poultry and fishery industries, via multifaceted approaches like ectopically expressed MSTN propeptide, dominant negative competitor, RNA interference, gene targeting and gene editing[1–4]. Among these methods, programmable nuclease-mediated gene editing proved to be robust to introduce random indels into the coding sequence of MSTN, thus disrupting its frame readability. However, indel-dependent gene editing is site-specific, rather than sequence-specific, and it introduces heterogenic mutations into double-stranded breaks (DSB). This makes the method unideal to gene-edited animal breeding because it requires multiple crossings for genotypically identical alleles. Therefore, it is desirable to develop a predictable and sequence-specific gene editing strategy for high-fidelity genetic engineering in farm animal research.

With the generation of MSTN-KO animals in recent years, MSTN's molecular function in lipid metabolism has gained increasing attention. It was observed that MSTN inactivation not only reduces overall body fat mass, but to some extent also the intramuscular fat (IMF) content in meat[5–7]. As IMF content is an important indicator of meat quality, its reduction would undermine the nutritional and economic value of double muscling animals. Therefore, a new tradeoff between augmented muscle growth and constant IMF is emerging as a high-priority breeding goal. In order to achieve this end, it is necessary to decode the molecular mechanism of MSTN's physiological role in adipogenesis and lipid metabolism.

In this study, we first devised an improved gene targeting strategy relying on dual fluorescence-assisted selection (DUFAS) to couple CRISPR/Cas9 with homology-mediated recombination (HDR) for two purposes: 1, to assure the precise mutation of MSTN, and 2, to enrich the on-target events with high efficiency. Using this new strategy, we generated cloned MSTN-KO Meishan pigs, which exhibited a typical double muscled phenotype. We then performed RNA-seq to profile transcriptomes of MSTN-KO and WT pigs and intersected the differentially expressed mRNAs and miRNAs to identify miRNA-mRNA regulatory pairs. We experimentally validated the existence of a novel cascade via which MSTN regulates fatty acid metabolism and fat deposition in pigs.

## Results

**DUFAS is robust to generate MSTN-KO pigs.** Targeted genomic engineering can be accomplished with gene editors like CRISPR/Cas9 via error-prone non-homologous end joining (NHEJ) repair pathway, but NHEJ-mediated gene editing is not predictable and tends to introduce heterogeneous mutations. HDR is error-free for mutation-specific gene editing, but it is technically challenging due to its low efficiency in mammalian cells. To overcome this bottleneck, we devise DUFAS that utilizes juxtaposed green and red fluorescence protein genes as reciprocal visually selectable markers to identify on-target HDR events by interrogating the fluorescence emitted from the isolated antibiotics-resistant cell clones. The presence of red fluorescence indicates random integration events, and only cell clones that are solely positive for enhanced green fluorescence protein (EGFP) are potential candidates for correctly targeted cell clones (Fig. 1a and Supplementary Fig. 1).

As a proof-of-principle test, we applied this strategy to three targeting sites at MSTN coding sequence (named T1, T2 and T3,

respectively) in the immortal cell line PK15. Three donor DNAs that have homologous arms immediately surrounding the 20 nt Cas9-editing sites were co-transfected with Cas9/gRNA plasmids into PK15 cell line for MSTN targeting (Fig. 1b). Junction PCR verified the correct targeting of the selectable marker gene into the three sites, respectively (Fig. 1c and Supplementary Fig. 2). Fluorescence observation showed that the targeted cell clones exhibited green fluorescence only (Fig. 1d), consistent with the principle of DUFAS (Supplementary Fig. 3). The targeting efficiency was 80.5%, 86.7%, and 82.3% at the three targeting sites, respectively (Supplementary Table 1). This test demonstrates that DUFAS is a robust strategy for the enrichment of on-target HDR events.

Meishan pig is an autochthonal pig breed in China and recognized as a fat-type pig due to low leanness. We aim to improve the leanness of Meishan pigs by disrupting MSTN. We then performed DUFAS-mediated gene targeting at the T1 site of the MSTN exon 1 in male Meishan pig primary fetal fibroblast cells and obtained an efficiency of 82.3% (Fig. 1e and Supplementary Table 1). One of the targeted cell clones (#MS30) was used for somatic cell nuclear transfer (SCNT). A total of 328 reconstructed embryos were transferred into 2 surrogate sows, one of which aborted and the other delivered 4 live male cloned piglets (Supplementary Table 2). These piglets were genotyped by Southern blotting to verify the insertion of the positive selection marker into one allele of the targeting site (Fig. 1f, g). These results proved that we successfully generated cloned MSTN-KO (MSTN$^{+/-}$) Meishan pigs.

Growth performance analysis showed that MSTN-KO Meishan pigs exhibited greater average daily weight gain (447.7 ± 10.25 g/d vs. 412.4 ± 10.10 g/d, $p = 0.0496$; Table 1) and more efficient feed conversion ratio (3.71 ± 0.07 vs. 4.2 ± 0.08, $p = 0.0037$; Table 1), compared to the WT controls (MSTN$^{+/+}$). Slaughter trials demonstrated that lean percentage of the MSTN-KO Meishan pigs was significantly improved (67.86 ± 1.32% vs. 49.40 ± 0.98%, $p < 0.0001$; Fig. 1h) over the WT. Consistent with these changes, the longissimus size and the backfat thickness were inversely altered, i.e. the former was significantly enhanced (43.39 ± 2.69 cm$^2$ vs. 31.17 ± 1.12 cm$^2$, $p = 0.0057$; Table 1) and the latter was significantly decreased (1.30 ± 0.21 cm vs. 2.49 ± 0.43 cm, $p = 0.0469$; Fig. 1i). Meat quality testing revealed that the content of IMF in MSTN-KO pigs decreased (1.28 ± 0.11% vs. 3.46 ± 0.16%, $p < 0.0001$; Fig. 1j, k and Table 1). Moreover, the decreased backfat thickness was associated with a reduced triglyceride (TG) content in the blood of the MSTN-KO pigs (0.28 ± 0.02 mmol/L vs. 0.44 ± 0.03 mmol/L, $p = 0.0029$; Table 1). Western blotting showed that MSTN expression decreased 56.23% compared to the WT controls, in line with the mono-allelic KO of MSTN (Fig. 1l). These analyses demonstrated that the MSTN-KO Meishan pigs exhibited a double muscling phenotype, thus forming the basis for the development of a new high-leanness pig line.

**Deep sequencing to dig out miRNA-mRNA pairs.** Having established that MSTN-KO Meishan pigs presented a pronounced double muscling phenotype with decreased IMF content, we set out to elucidate how MSTN regulates the fat deposition, and how to manipulate this mechanism in a way to increase muscle mass while maintaining a reasonable IMF content. To address this question, we performed mRNA and miRNA deep sequencing of subcutaneous fat tissue samples from the MSTN-KO and WT pigs in order to identify miRNA-mRNA regulatory pairs potentially involved in the fat deposition (Fig. 2a).

The mRNA sequencing produced a total of 70,632,750 and 64,526,793 clean reads on average in the two mRNA libraries.

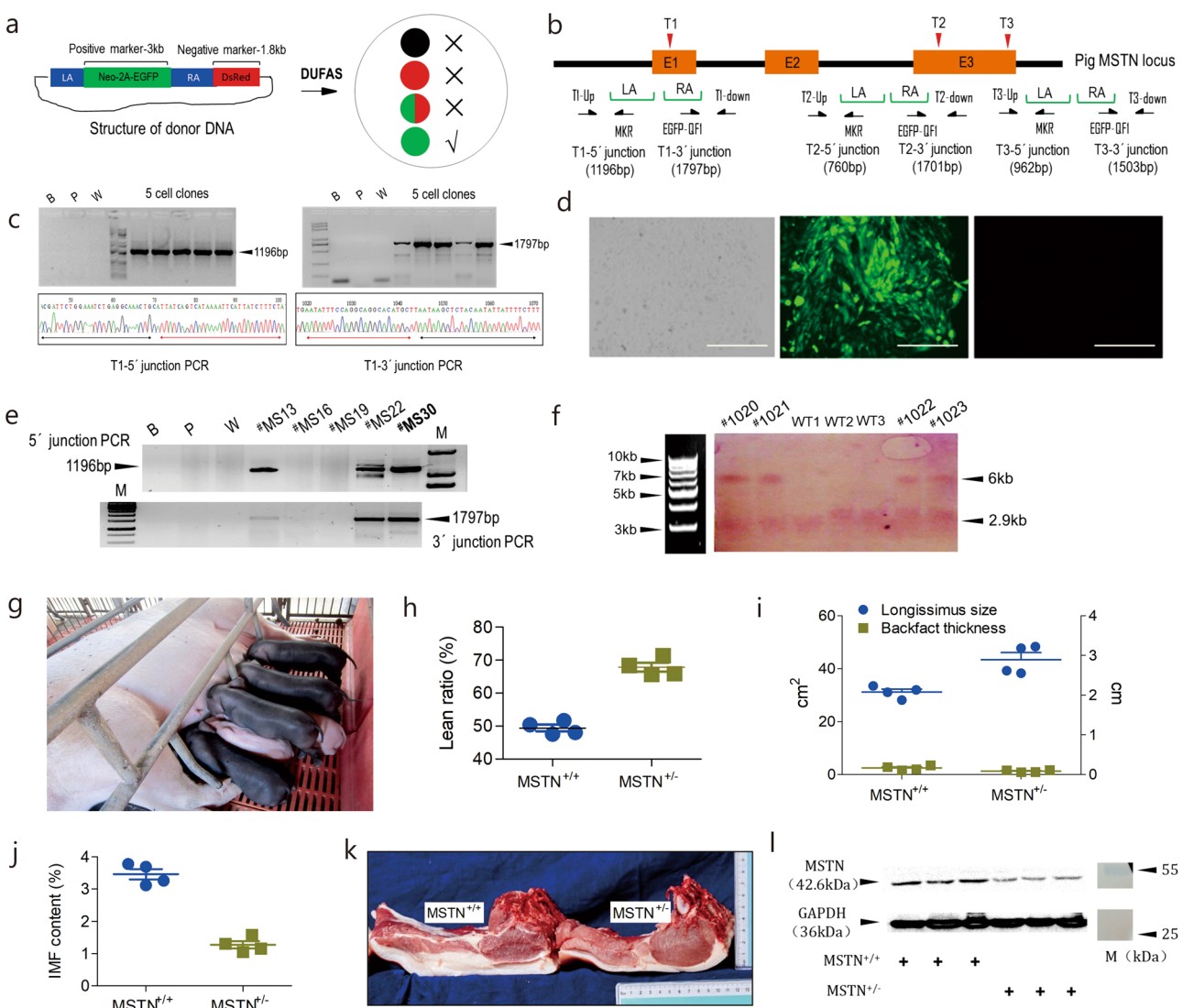

**Fig. 1 Generation and phenotyping of cloned MSTN-KO Meishan pigs via DUFAS. a** Schematic of DUFAS-mediated HDR. LA, left arm. RA, right arm. "×", non-targeted cell clones. "√", targeted cell clones. **b** Outline of porcine MSTN targeting by DUFAS. LA and RA, as stated above. E1, E2, and E3 are the exons of MSTN. T1, T2, and T3 denote three CRISPR/Cas9 targeting sites. **c** Results of the junction PCR of targeted cell clones at the T1 site. B blank control; P negative control templated by donor DNA; W negative control templated by WT genomic DNA. Sanger sequencing results of the junction PCR are shown at the bottom. Black and red arrowheads represent the genomic sequence and homologous arm sequence, respectively. **d** Representative image of a targeted cell clone by DUFAS. Left, bright field. Middle, EGFP emission. Right, DsRed emission. Scale bar, 10 μm. **e** PCR of DNA insertion at the T1 site in the primary fetal fibroblast cells of Meishan pigs. B, P and W, defined as above. #MS16, #MS19, #MS13, #MS22, and #MS30 are five candiadte cell clones. M is 100 bp DNA ladder. **f** Genotyping of the newborn piglets by Southern blotting. WT wild-type genomic DNA as blotting substrate. WT1, WT2 and WT3 are three biological repeats. Black triangles denote the length of the blotted DNA fragments. The four piglets are numbered as #MS1020, #MS1021, #MS1022, and #MS1023. M is 1 kb DNA ladder (N3232L, NEB). **g** Image of the four cloned MSTN-KO Meishan piglets. **h** Lean ratio of MSTN$^{+/+}$ and MSTN$^{+/-}$ pigs ($n = 4$ biologically independent animals). **i** Longissimus size and backfat thickness of MSTN$^{+/+}$ and MSTN$^{+/-}$ pigs ($n = 4$ biologically independent animals). **j** IMF content of MSTN$^{+/+}$ and MSTN$^{+/-}$ pigs ($n = 4$ biologically independent animals). **k** Transverse section of the longissimus from MSTN$^{+/+}$ and MSTN$^{+/-}$ pigs. **l** MSTN expression by Western blotting in MSTN$^{+/+}$ and MSTN$^{+/-}$ pigs. GAPDH is the internal control. M is the Page Ruler Prestained Protein Ladder 26616 (ThermoFisher Scientific).

Among these, 64,207,565 (90.9% of clean reads) and 57,943,539 (89.8% of clean reads), respectively, were mappable and could be aligned to unique mRNAs. The transcript coverage was 116.32× and 104.56× on average, respectively (Supplementary Table 3). Among the 18,877 transcripts detected, 4524 differentially expressed mRNAs (DE mRNA; $p < 0.05$ and $|\log_2(\text{Fold change})| > 1$) were identified based on fragments per kilobase million (FPKM) > 10 in either of the two libraries. Among these DE mRNAs, 3059 were up-regulated and 1465 were down-regulated (Fig. 2b and Supplementary Fig. 4). The deep sequencing results were verified

by quantitative real-time PCR (qPCR) for randomly selected 10 upregulated and 10 downregulated DE mRNAs using the total RNA isolated from the subcutaneous fat tissues. Results of qPCR assays of the 20 DE mRNAs were in agreement with deep sequencing results in terms of expression alteration (up/down) and statistical significance (Supplementary Figs. 6 and 7).

The small RNA sequencing produced a total of 10,623,793 and 9,009,158 clean reads on average in the two small RNA libraries, among which 8,893,765 (83.72% of clean reads) and 6,321,386 (70.17% of clean reads) were mappable and could be

**Table 1 Phenotypic analysis of cloned MSTN-KO Meishan pigs.**

| Phenotype index | MSTN$^{+/+}$ ($n = 3$) | MSTN$^{+/-}$ ($n = 3$) |
|---|---|---|
| Growth performance | | |
| ADWG (g/d) | 412.4 ± 10.10[a] | 447.7 ± 10.25[b] |
| FCR | 4.2 ± 0.08[a] | 3.71 ± 0.07[b] |
| Carcass performance | | |
| Longissimus size (cm$^2$) | 31.17 ± 1.12[a] | 43.39 ± 2.69[b] |
| Backfat thickness (cm) | 2.49 ± 0.43[a] | 1.30 ± 0.21[b] |
| Lean ratio (%) | 49.40 ± 0.98[a] | 67.86 ± 1.32[b] |
| Meat quality | | |
| IMF (%) | 3.46 ± 0.16[a] | 1.28 ± 0.11[b] |
| Tenderness (N) | 68.10 ± 3.99[a] | 77.55 ± 3.28[b] |
| Hematology | | |
| LDL (mmol/L) | 1.52 ± 0.08[a] | 0.84 ± 0.07[b] |
| TG (mmol/L) | 0.44 ± 0.03[a] | 0.28 ± 0.02[b] |

Values with different superscripts in the same row are considered significantly different. ADWG average daily weight gain, FCR feed conversion ratio, IMF intramuscular fat, LDL low-density lipoprotein, TG triglyceride.

aligned to unique sequences in the porcine genome (Supplementary Table 4). Except rRNAs, tRNAs, snRNAs and snoRNAs (based on Rfam 11.0), a total of 245 and 253 known miRNAs were identified from the two small RNA libraries, respectively, according to the mature miRNA sequences in the miRBase. Among these miRNAs, 70 differentially expressed miRNAs (DE miRNA; $p < 0.05$ and |log$_2$(Fold change)|>1) were identified on the basis of reads per million (RPM) > 5 criterion in either of the two libraries, including 40 upregulated and 30 downregulated DE miRNAs (Fig. 2c and Supplementary Fig. 5). Similarly, the deep sequencing results were also verified by qPCR for 20 randomly selected DE miRNAs (10 upregulated and 10 downregulated) using total RNA isolated from the subcutaneous fat tissues. The expression of the 20 DE miRNAs showed consistency between qPCR assays and deep sequencing results in terms of expression alteration (up/down) and statistical significance (Supplementary Figs. 8 and 9).

We then aimed to identify miRNA-mRNA regulatory pairs by intersecting the DE mRNA and miRNA. The target genes for the DE miRNAs were predicted on the basis of porcine genome sequence using the miRanda algorithms. A total of 2615 target genes for the 70 DE miRNAs were identified, of which 388 target genes were DE mRNAs. We then intersected these 388 target DE mRNAs against DE miRNAs that have inversely correlated expression, i.e. upregulated DE miRNAs were paired with downregulated predicted target DE mRNAs and downregulated DE miRNAs were paired with upregulated predicted target DE mRNAs. This generated 73 miRNA:mRNA regulatory pairs (Table 2). GO analysis of these intersected DE mRNAs showed that 15 GO terms were enriched in the biological processes, including the biological adhesion and cell migration; 15 GO terms were enriched in cellular components, mainly in the cell and organelle membrane; 15 GO terms were enriched in molecular functions, for example lipid metabolism (Fig. 2d). KEGG analysis showed that the enrichment mainly occurred in signal pathways such as PI3K-Akt and TGF-β (Fig. 2e). To better understand the interaction between the intersected DE mRNAs and miRNAs, we constructed an miRNA-mRNA network (Supplementary Fig. 10). By analyzing the five hub node genes THY1, THBS1, ITGB3, SCD5, and BMPR15, we found that stearoyl-CoA desaturase 5 (SCD5) was the only gene whose molecular role could be directly linked to fat deposition in pigs. SCD5 is an integral membrane protein of the endoplasmic reticulum that catalyzes the formation of monounsaturated fatty acids from saturated fatty acids, in which the monounsaturated fatty acids would be the substrates for de novo triglyceride biological synthesis[8]. Its function prompts us to hypothesize that SCD5 plays a role in fat deposition and its regulation affects the phenotype in MSTN-KO pigs.

**miR222 suppresses SCD5 to reduce nonsaturated fatty acids.** As we predicted SCD5 to be regulated by five miRNAs (miR370, miR222, miR2483, miR141, and miR486) in the miRNA-mRNA regulatory pair list, we first tested which miRNA could target SCD5 to suppress its expression. Subcutaneous preadipocytes were treated with mimics of these five miRNA, and SCD5 mRNA abundance was quantified by qPCR. This showed that SCD5 mRNA was decreased only by the miR222 mimic treatment, while the other four miRNA mimics failed to suppress SCD5 (Fig. 2f).

qPCR and Northern blotting were conducted to validate the expression of miR222 in subcutaneous fat tissue, which showed that miR222 was upregulated in MSTN-KO pigs, compared to that in the WT pigs (Fig. 2g, h). qPCR and Western blotting were performed on the subcutaneous fat tissues to quantify the expression of SCD5, which showed that SCD5 was downregulated at both mRNA and protein levels (Fig. 2i, j). These results demonstrate the inverse correlation between miR222 and SCD5 expression, implying that SCD5 is likely to be targeted by miR222.

We next analyzed the evolutionary conservation of miR222 in 15 animal species. This analysis revealed that miR222 was highly evolutionary conserved as its seed region had 100% identity and its mature sequence also had nearly 100% identity (Fig. 2k). We further inspected the potential binding positions of miR222 in the 3′UTR of SCD5 in five vertebrate species (pig, sheep, cattle, human and chicken), which revealed that the interaction between the seed region of miR222 with the SCD5 3′UTR at base from 1507 to 1513 was also highly conserved (Fig. 2l). This suggests that the biological function of miR222 is evolutionarily conserved. We were inspired by this notion to study the putative regulatory effect of miR222 on SCD5.

To achieve this, subcutaneous preadipocytes were treated by miR222 mimics and inhibitors and SCD5 expression was examined by qPCR and Western blotting. SCD5 mRNA abundance and protein levels were inversely correlated with miR222 overexpression and inhibition, suggesting that endogenous SCD5 was targeted by miR222 (Fig. 3a, b and Supplementary Fig. 11). The dual luciferase reporter test showed that miR222 mimics suppressed SCD5 with the wild-type 3′UTR, and that this suppression was nullified with the mutant 3′UTR (Fig. 3c, d). The above two assays confirmed that miR222 can directly target the 3′ UTR of SCD5 to suppress its expression.

We next sought to analyze the physiological consequence of miR222's regulation on SCD5. As stated previously, SCD5 is a Δ9 fatty acid desaturase that catalyzes the conversion from saturated to monounsaturated fatty acids, where the catalyzed products would be substrates for the TG synthesis. This gene preferentially desaturates two saturated fatty acids, palmic C16:0 and stearic C18:0 acids, into monounsaturated fatty acids palmitoleic C16:1 and oleic C18:1 acids. On account of this, we first quantified the TG content of subcutaneous preadipocytes treated with miR222 mimics and inhibitors. As shown in Fig. 3e, miR222 mimics treatment reduced the TG content, whereas miR222 inhibitor increased it. Secondly, we profiled the fatty acid spectrum and found that the ratio of C16:1 to C16:0 and C18:1 to C18:0 were inversely correlated with miR222 abundance and positively correlated with SCD5 expression (Fig. 3f). These tests suggested that the fatty acid spectrum alteration was a consequence of miR222's regulation on SCD5.

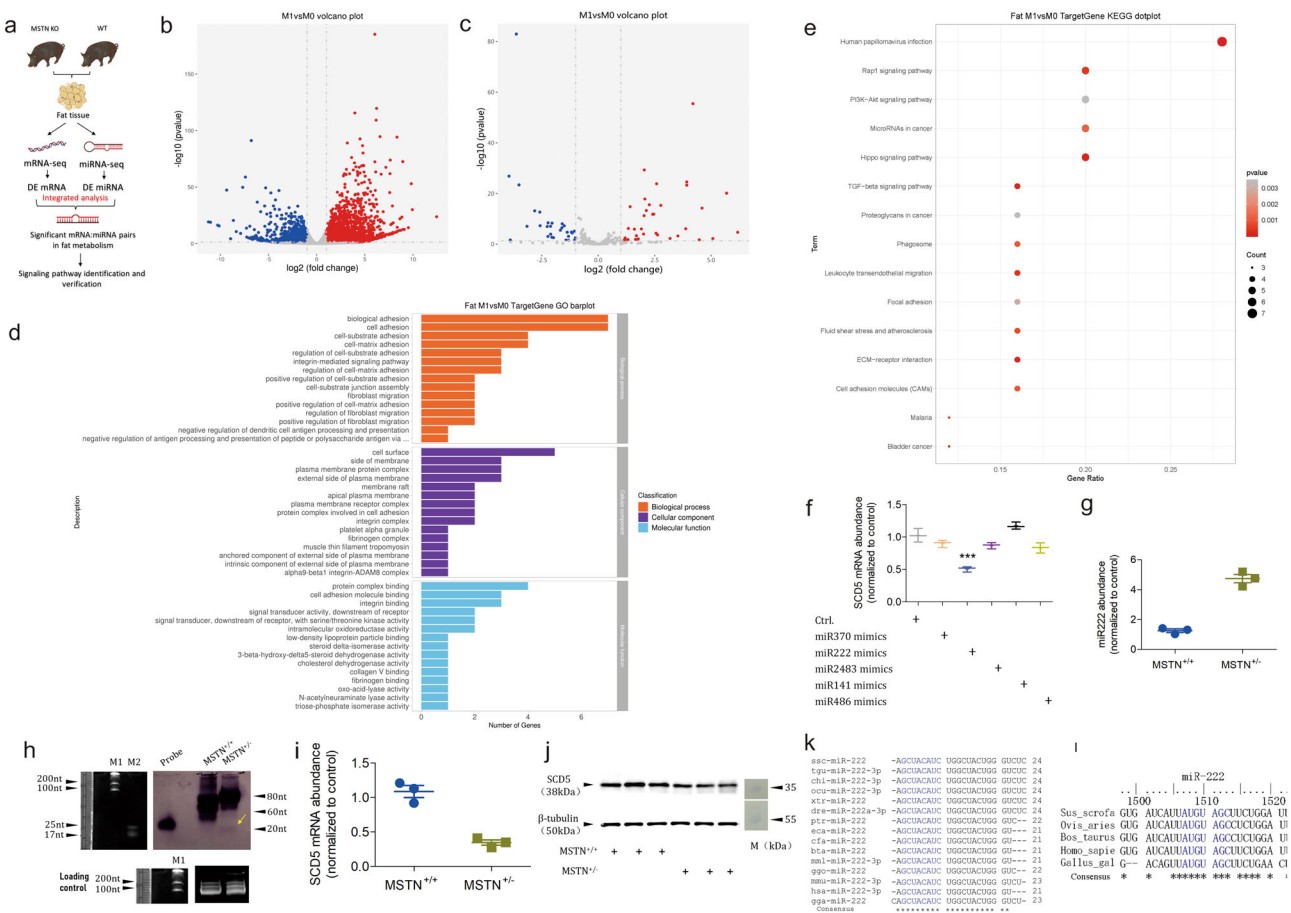

**Fig. 2 Identification of miRNA-mRNA regulatory pairs potentially involved in fat deposition. a** The schematic of mRNA and miRNA deep sequencing and bioinformatic analysis aimed at identification of potential signaling pathways. **b** Volcano plot of the differentially expressed mRNAs between MSTN-KO and WT pigs. **c** Volcano plot of the differentially expressed miRNAs between MSTN-KO and WT pigs. M0 and M1 represent WT and MSTN-KO pigs, respectively. **d** GO barplot of differentially expressed mRNAs involved in lipid metabolism. **e** KEGG dotplot of differentially expressed mRNAs involved in lipid metabolism. **f** qPCR analysis of SCD5 mRNA abundance in response to treatments with five miRNAs. **g** qPCR analysis of miR222 abundance in the subcutaneous fat tissue of WT and MSTN-KO pigs ($n = 3$ biologically independent samples, respectively). **h** Northern blotting of miR222 in WT and MSTN-KO pigs. The yellow arrowhead denotes the position of miR222. Small RNAs were loaded in equal amounts. Probe against miR222 was used as positive control for DIG colorization. M1 and M2 are RiboRuler Low Range RNA Ladder (SM1831, ThermoFisher Scientific) and microRNA Marker (N2102, NEB). **i** qPCR analysis of endogenous SCD5 abundance in the subcutaneous fat tissue of WT and MSTN-KO pigs ($n = 3$ biologically independent samples, respectively). **j** Western blotting of the endogenous SCD5 in the subcutaneous fat tissue of WT and MSTN-KO pigs ($n = 3$ biologically independent samples, respectively). M is the Page Ruler Prestained Protein Ladder 26616 (ThermoFisher Scientific). **k** Evolutionary conservation of miR222 in multiple species. Red letters represent the seed sequence of miR222. "*" Represents the evolutionary consensus sequence of miR222. **l** Binding sites of miR222 on the 3′UTR of SCD5 in five animal species. Red letters represent the annealing region of seed sequence of miR222. "*" Shows the consensus sequence of 3′UTR of SCD5 in the five animal species.

Furthermore, we tested the TG content and fatty acid spectrum in the subcutaneous fat tissue and preadipocytes of the MSTN-KO and wild-type pigs, without treatment of exogenous nucleic acid. TG content and the ratio of C16:1 to C16:0 and C18:1 to C18:0 concomitantly decreased in both the subcutaneous fat tissue and preadipocytes (Fig. 3g–j). In agreement with the decreased TG content, Oil Red O staining of the differentiated subcutaneous preadipocytes showed less lipid drops in the MSTN-KO pigs (Fig. 3k).

Taken together, the findings above suggested that SCD5 is an endogenous molecular target of miR222, and the regulatory effect of miR222 on SCD5 impacts the fatty acid composition and TG content.

**MEF2C acts downstream of MSTN to regulate miR222.** We next sought to find out how miR222 was regulated. From the

perspective of posttranscriptional bioprocessing, we analyzed the key enzymes involved in the miRNA processing pathway by qPCR and found that the expression of Drosha and Dicer remained unchanged (Supplementary Figs. 12 and 13). This ruled out the possibility that miR222 was post-transcriptionally processed to alter its abundance. We then looked for transcription factors (TF) that might transcriptionally affect the expression of miR222. Using the web server PROMO, we predicted a list of 44 TFs that were likely to bind to the promoter region of miR222. By intersecting this TF list with the RNA-seq data, we filtered out 38 non-significantly differential TFs; among the remaining six TFs significantly upregulated with MSTN-KO, MEF2C was the top-ranked one (Supplementary Table 5). Two potential binding sites (site A and B) on the miR222 promoter were predicted to be bound by MEF2C. We next performed several lines of assays to prove whether or not miR222 was transcriptionally regulated by MEF2C.

**Table 2 miRNA-mRNA regulatory pairs identified by bioinformatics prediction and RNA-seq analysis.**

| DE miRNA | | | DE mRNA | | | |
|---|---|---|---|---|---|---|
| Name | Log$_2$FC | *p*-value | Name | Log$_2$FC | *p*-value | mRNA description |
| ssc-miR-370 | 6.178595091 | 2.07828E-05 | SCD5 | −1.237577 | 0.009508 | Stearoyl-CoA desaturase 5 |
| ssc-miR-370 | | | ITGB1BP2 | −1.472781 | 0.014478 | Integrin subunit beta 1 binding protein 2 |
| ssc-miR-370 | | | BMPR1B | −1.11218 | 2.13E-06 | Bone morphogenetic protein receptor type 1B |
| ssc-miR-218b | 5.689812352 | 3.25407E-21 | THBS1 | −2.471627 | 1.02E-24 | Thrombospondin-1 |
| ssc-miR-218 | 5.688618566 | 3.02648E-21 | THBS1 | −2.471627 | | Thrombospondin-1 |
| ssc-miR-218-5p | 5.688618566 | 3.02648E-21 | THBS1 | −2.471627 | | Thrombospondin-1 |
| ssc-miR-758 | 4.596925271 | 5.84363E-15 | FGL2 | −1.910608 | 0.005372 | Fibrinogen like 2 |
| ssc-miR-758 | | | TPM1 | −1.50469 | 7.95E-16 | Tropomyosin 1 |
| ssc-miR-758 | | | BMPR1B | −1.11218 | 2.13E-06 | Bone morphogenetic protein receptor type 1B |
| ssc-miR-381-3p | 4.19198935 | 2.68577E-56 | PECAM1 | −1.015514 | 0.044941 | Platelet endothelial cell adhesion molecule 1 |
| ssc-miR-9-1 | 3.922280627 | 2.01609E-24 | BMPR1B | −1.11218 | 2.13E-06 | Bone morphogenetic protein receptor type 1B |
| ssc-miR-9-2 | 3.922280627 | 2.01609E-24 | BMPR1B | −1.11218 | | Bone morphogenetic protein receptor type 1B |
| ssc-miR-9 | 3.922280627 | 2.01609E-24 | BMPR1B | −1.11218 | | Bone morphogenetic protein receptor type 1B |
| ssc-miR-133a-5p | 3.875394952 | 7.87582E-07 | ITGB3 | −2.86968 | 1.99E-20 | Integrin subunit beta 3 |
| ssc-miR-326 | 3.528143119 | 0.000584564 | TNC | −2.471314 | 0.009831 | Tenascin C |
| ssc-miR-326 | | | BMPR1B | −1.11218 | 2.13E-06 | Bone morphogenetic protein receptor type 1B |
| ssc-miR-326 | | | ITGB3 | −2.86968 | 1.99E-20 | Integrin subunit beta 3 |
| ssc-miR-326 | | | FMOD | −4.940557 | 5.15E-18 | Fibromodulin |
| ssc-miR-222 | 2.522752887 | 7.57707E-31 | THBS1 | −2.471627 | 1.02E-24 | Thrombospondin-1 |
| ssc-miR-222 | | | SCD5 | −1.237577 | 0.009508 | Stearoyl-CoA desaturase 5 |
| ssc-miR-222 | | | BMPR1B | −1.11218 | 2.13E-06 | Bone morphogenetic protein receptor type 1B |
| ssc-miR-2483 | 2.460120488 | 1.861E-12 | SCD5 | −1.237577 | 0.009508 | Stearoyl-CoA desaturase 5 |
| ssc-miR-708-5p | 2.129951111 | 3.82258E-14 | PPP2R1B | −1.18985 | 2.41E-11 | Protein phosphatase 2, regulatory subunit A, beta |
| ssc-miR-708-5p | | | PECAM1 | −1.015514 | 0.044941 | Platelet endothelial cell adhesion molecule 1 |
| ssc-miR-210 | 2.062574633 | 1.46791E-18 | BMPR1B | −1.11218 | 2.13E-06 | Bone morphogenetic protein receptor type 1B |
| ssc-miR-708-3p | 2.039435793 | 3.15227E-30 | GBP1 | −3.500134 | 0.043545 | Guanylate-binding protein-1 |
| ssc-miR-139-3p | 2.004837087 | 9.58703E-05 | PPP2R1B | −1.18985 | 2.41E-11 | Protein phosphatase 2, regulatory subunit A, beta |
| ssc-miR-204 | 1.735135479 | 0.011872702 | XIRP2 | −4.581904611 | | |
| ssc-miR-204 | | | FMOD | −4.940557268 | 5.15E-18 | Fibromodulin |
| ssc-miR-145-5p | 1.647812344 | 0.012650534 | TNC | −2.471314 | 0.009831 | Tenascin C |
| ssc-miR-145-5p | | | PPP2R1B | −1.18985 | 2.41E-11 | Protein phosphatase 2, regulatory subunit A, beta |
| ssc-miR-139-5p | 1.332128509 | 6.31401E-07 | PPP2R1B | −1.18985 | 2.41E-11 | Protein phosphatase 2, regulatory subunit A, beta |
| ssc-miR-2366 | 1.317737461 | 0.014450992 | BMPR1B | −1.11218 | 2.13E-06 | Bone morphogenetic protein receptor type 1B |
| ssc-miR-2366 | | | PPP2R1B | −1.18985 | 2.41E-11 | Protein phosphatase 2, regulatory subunit A, beta |
| ssc-miR-2366 | | | TPM1 | −1.50469 | 7.95E-16 | Tropomyosin 1 |
| ssc-miR-2366 | | | PECAM1 | −1.015514 | 0.044941 | Platelet endothelial cell adhesion molecule 1 |
| ssc-miR-2366 | | | BMPR1B | −1.11218 | 2.13E-06 | Bone morphogenetic protein receptor type 1B |
| ssc-miR-2366 | | | ITGB3 | −2.86968 | 1.99E-20 | Integrin subunit beta 3 |
| ssc-miR-141 | 1.186293739 | 0.003239735 | SCD5 | −1.237577 | 0.009508 | Stearoyl-CoA desaturase 5 |
| ssc-miR-486 | 1.157635382 | 0.01879132 | TPM1 | −1.50469 | 7.95E-16 | Tropomyosin 1 |
| | | | SCD5 | −1.237577 | 0.009508 | Stearoyl-CoA desaturase 5 |
| ssc-miR-455-3p | -3.9566311 | 1.55911E-27 | TSPO | 2.0494085 | 1.28E-22 | Translocator protein |
| ssc-miR-455-5p | -3.63262127 | 1.03633E-83 | CLIC5 | 2.0277318 | 0.014776 | Chloride intracellular channel 5 |

**Table 2 (continued)**

| DE miRNA | | | DE mRNA | | | |
|---|---|---|---|---|---|---|
| Name | Log$_2$FC | p-value | Name | Log$_2$FC | p-value | mRNA description |
| ssc-miR-455-5p | | | ADM | 1.5099872 | 0.0019 | Adrenomedullin receptor binding |
| ssc-miR-503 | -2.58773042 | 1.25655E-13 | UPK3A | 2.3662439 | 0.000114 | Uroplakin-3a |
| ssc-miR-450b-3p | -2.11042317 | 0.000111361 | SERPINE1 | 1.8087562 | 0.023176 | Serine protease inhibitor |
| ssc-miR-542-5p | -2.05471325 | 0.001828484 | SLA-1 | 2.2493011 | 7.35E-13 | Swine leukocyte antigen 1 |
| ssc-miR-33b-3p | -1.43038729 | 0.022697288 | NOTCH4 | 1.5662127 | 8.76E-05 | Neurogenic locus notch homolog protein 4 |
| ssc-miR-33b-3p | | | CLIC5 | 2.0277318 | 0.014776 | Chloride intracellular channel 5 |
| ssc-miR-33b-3p | | | HOPX | 1.5568756 | 1.76E-07 | Homeodomain only protein x |
| ssc-miR-33b-3p | | | CLIC5 | 2.0277318 | 0.014776 | Chloride intracellular channel 5 |
| ssc-miR-30c-1-3p | -1.38003544 | 1.60307E-05 | NPL | 1.2288895 | 4.61E-08 | N-acetylneuraminate pyruvate lyase |
| ssc-miR-30c-1-3p | | | CLIC5 | 2.0277318 | 0.014776 | Chloride intracellular channel 5 |
| ssc-miR-30c-1-3p | | | BGN | 6.0384272 | 6.5E-186 | Biglycan |
| ssc-miR-187 | -1.31451094 | 0.027568031 | CEBPA | 2.6207539 | 2.75E-12 | CCAAT Enhancer Binding Protein alpha |
| ssc-miR-4331-5p | -1.19068004 | 1.75792E-05 | ITGB2 | 2.3291911 | 1.11E-10 | Integrin subunit beta 2 |
| ssc-miR-107 | -1.060541296 | 5.86371E-06 | MAPK12 | 1.376323493 | 2.06E-09 | Mitogen-activated protein kinase 12 |
| ssc-miR-107 | | | TPI1 | 1.010227085 | 8.48E-08 | Triosephosphate isomerase 1 |
| ssc-miR-107 | | | ITGB2 | 2.329191133 | 1.11E-10 | Integrin subunit beta 2 |
| ssc-miR-4334-3p | -1.048828923 | 0.019963139 | CDH1 | 1.183998589 | 1.61E-08 | Cadherin 1 |
| ssc-miR-4334-3p | | | SLC26A11 | 1.615163318 | 4.34E-06 | Solute carrier family 26 member 11 |
| ssc-miR-4334-3p | | | THY1 | 1.409988419 | 0.001799 | Thy-1 cell surface antigen |
| ssc-miR-4334-3p | | | COL1A1 | 1.706614176 | 1.21E-09 | Collagen type I alpha 1 chain |
| ssc-miR-4334-3p | | | SERPINE1 | 1.808756158 | 0.023188 | Serpin family E member 1 |
| ssc-miR-532-3p | -1.012120672 | 0.002619262 | SLC16A3 | 1.260691201 | 2.33E-11 | Solute carrier family 16 member 3 |
| ssc-miR-532-3p | | | COL1A1 | 1.706614176 | 1.21E-09 | Collagen type I alpha 1 chain |
| ssc-miR-532-3p | | | BGN | 6.03842722 | 6.5E-186 | Biglycan |
| ssc-miR-532-3p | | | FLVCR2 | 4.988102404 | 0.000774 | Feline leukemia virus subgroup C cellular receptor family member 2 |
| ssc-miR-532-3p | | | CLIC5 | 2.0277318 | 0.014776 | Chloride intracellular channel 5 |
| ssc-miR-532-3p | | | CEBPA | 2.6207539 | 2.75E-12 | CCAAT Enhancer Binding Protein alpha |
| ssc-miR-532-3p | | | TPI1 | 1.010227085 | 8.48E-08 | Triosephosphate isomerase 1 |
| ssc-miR-532-3p | | | THY1 | 1.409988419 | 0.001799 | Thy-1 cell surface antigen |

Firstly, the MEF2C expression level was validated by qPCR and immunoblotting to corroborate the RNA-seq data. As shown in Fig. 4a, b., the MEF2C expression increased in the MSTN-KO samples, which is consistent with the transcriptomic result. Secondly, we examined if endogenous miR222 expression was affected by overexpression and silencing of MEF2C. The overexpression of the Flag-tagged MEF2C in subcutaneous preadipocytes significantly enhanced the miR222 abundance ($p = 0.0002$; Fig. 4c, d). Conversely, silencing of MEF2C with small interfering RNA (siRNA) significantly, reduced the miR222 abundance ($p = 0.0013$; Fig. 4e, f). These results suggested that the expression of miR222 and MEF2C is positively correlated. Thirdly, we used a dual luciferase reporter assay to detect whether or not specific DNA sequences of the miR222 promoter could be targeted by MEF2C (Fig. 4g). After

overexpressing the Flag-tagged MEF2C in subcutaneous pre-adipocytes, reporter plasmid with wild-type miR222 promoter sequence (pGL3-miR222-WT) had higher luciferase intensity than the control, whereas the reporter plasmid with the mutant miR222 promoter sequence (pGL3-miR222-Mut) had significantly decreased luciferase activity(Fig. 4h). We also found that the luciferase activity of pGL3-miR222-WT decreased with the silencing of MEF2C (Fig. 4i). These results suggest that MEF2C targets to specific region of the miR222's promoter to regulate its expression.

In order to validate whether or not the specific DNA sequence of the miR222 promoter could be directly bound by MEF2C, chromatin immunoprecipitation (ChIP) and electrophoretic mobility shift assay (EMSA) were carried out to examine their direct interaction. ChIP-PCR showed that the site A DNA

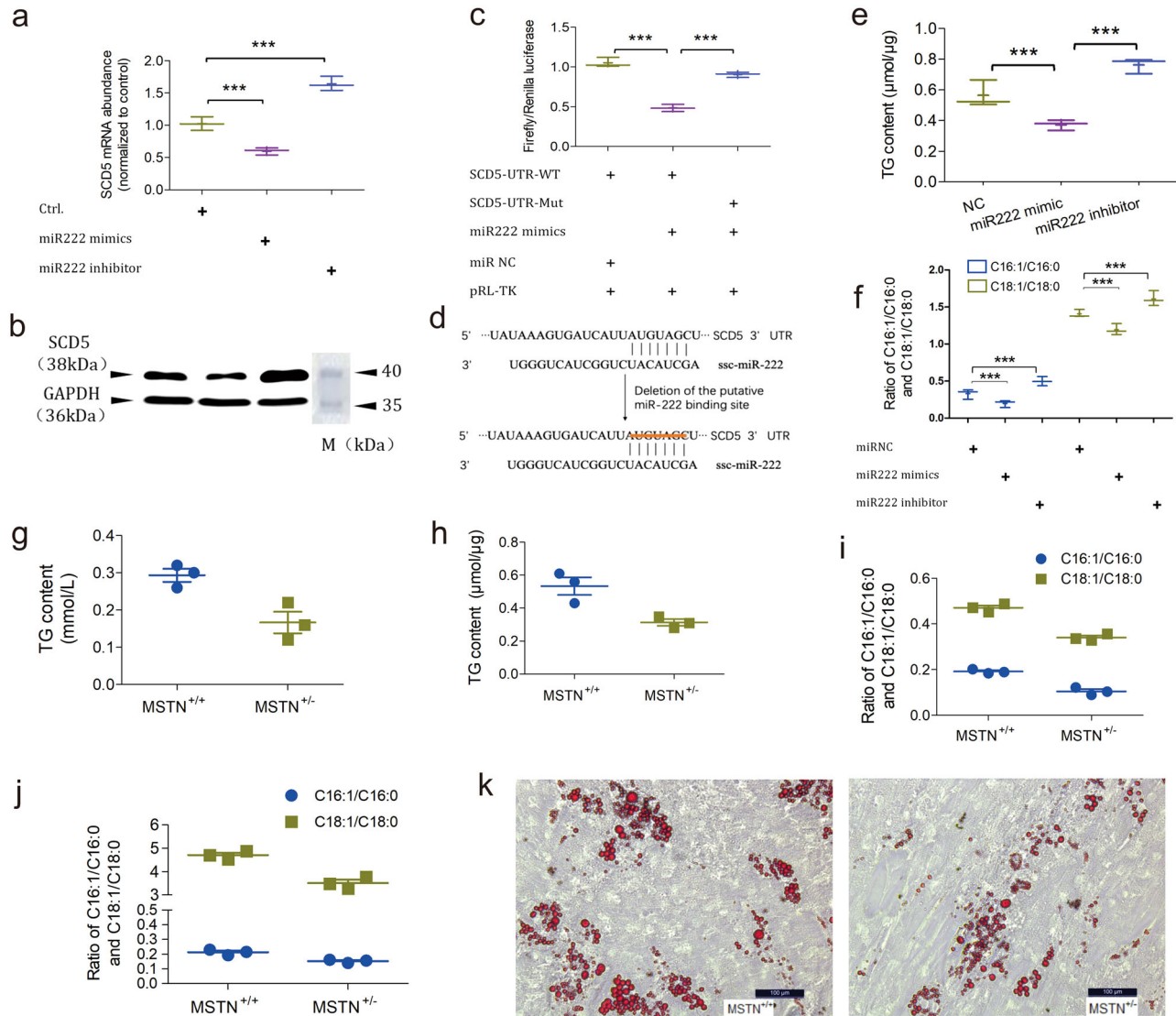

**Fig. 3 miR-222 inhibits SCD5 expression to regulate the fatty acid desaturation and TG content. a** qPCR of endogenous SCD5 mRNA abundance in response to miR222 overexpression and inhibition. $n = 3$ independent experiments. "***" Indicates statistically significant. **b** Western blotting of SCD5 protein level in response to miR222 overexpression and inhibition. GAPDH is the internal control. M is the Page Ruler Prestained Protein Ladder 26616 (ThermoFisher Scientific). **c** Dual luciferase reporter testing of miR222 on SCD5 3′UTR. pRL-TK is the *Renilla* luciferase expression plasmid used to normalize luciferase activity. $n = 3$ independent experiments. "***" Indicates statistically significant. **d** The binding site of miR222 in the SCD5 3′UTR and the introduced mutation. **e** TG content alteration in response to miR222 overexpression and inhibition in subcutaneous preadipocytes. $n = 3$ independent experiments. "***" Indicates statistically significant. **f** Alteration of C16:1/C16:0 and C18:1/C18:0 fatty acid ratio in subcutaneous preadipocytes in response to treatment with miR222 mimics and inhibitor. $n = 3$ independent experiments. "***" Indicates statistically significant. **g** TG content in the subcutaneous fat tissue of WT (MSTN$^{+/+}$) and MSTN-KO (MSTN$^{+/-}$) pigs ($n = 3$ biologically independent animals). **h** TG content in the subcutaneous preadipocytes of WT (MSTN$^{+/+}$) and MSTN-KO (MSTN$^{+/-}$) pigs ($n = 3$ biologically independent animals). **i** C16:1/C16:0 and C18:1/C18:0 fatty acid ratio in the subcutaneous fat tissue of WT (MSTN$^{+/+}$) and MSTN-KO (MSTN$^{+/-}$) pigs ($n = 3$ biologically independent animals). **j** C16:1/C16:0 and C18:1/C18:0 fatty acid ratio of subcutaneous preadipocytes of WT (MSTN$^{+/+}$) and MSTN-KO (MSTN$^{+/-}$) pigs ($n = 3$ biologically independent animals). **k** Representative image of Oil Red O staining of the subcutaneous preadipocytes of WT (MSTN$^{+/+}$) and MSTN-KO (MSTN$^{+/-}$) pigs. Scale bar, 100 μm.

sequence was successfully pulled down by the MEF2C antibody, whereas the site B was not pulled down (Fig. 4j). Sanger sequencing of the site A PCR product revealed that it was identical to the DNA sequence of miR222 promoter (Fig. 4k). This result was further confirmed by quantitative PCR of the pulldowns (Fig. 4l). This result suggests that MEF2C binds to site A, rather than site B of the miR222 promoter. Next, EMSA was performed with the nuclear extract of subcutaneous preadipocytes. A band shifting was observed when the biotin-labelled site A DNA fragment was incubated with the nuclear extract (the far right lane in the Fig. 4m). A super shifting band was observed

when MEF2C antibody was added to the reaction of biotin-labelled site A DNA fragment with nuclear extract (the second lane from the right in the Fig. 4m). Together, these results prove that miR222 is transcriptionally regulated by MEF2C through its direct binding to the given DNA sequence of the miR222 promoter.

**MSTN acts via MEF2C/miR222/SCD5 to affect fat deposition.** In order to validate that MSTN-mediated fatty acid desaturation is MEF2C/miR222/SCD5 cascade-dependent, we transiently

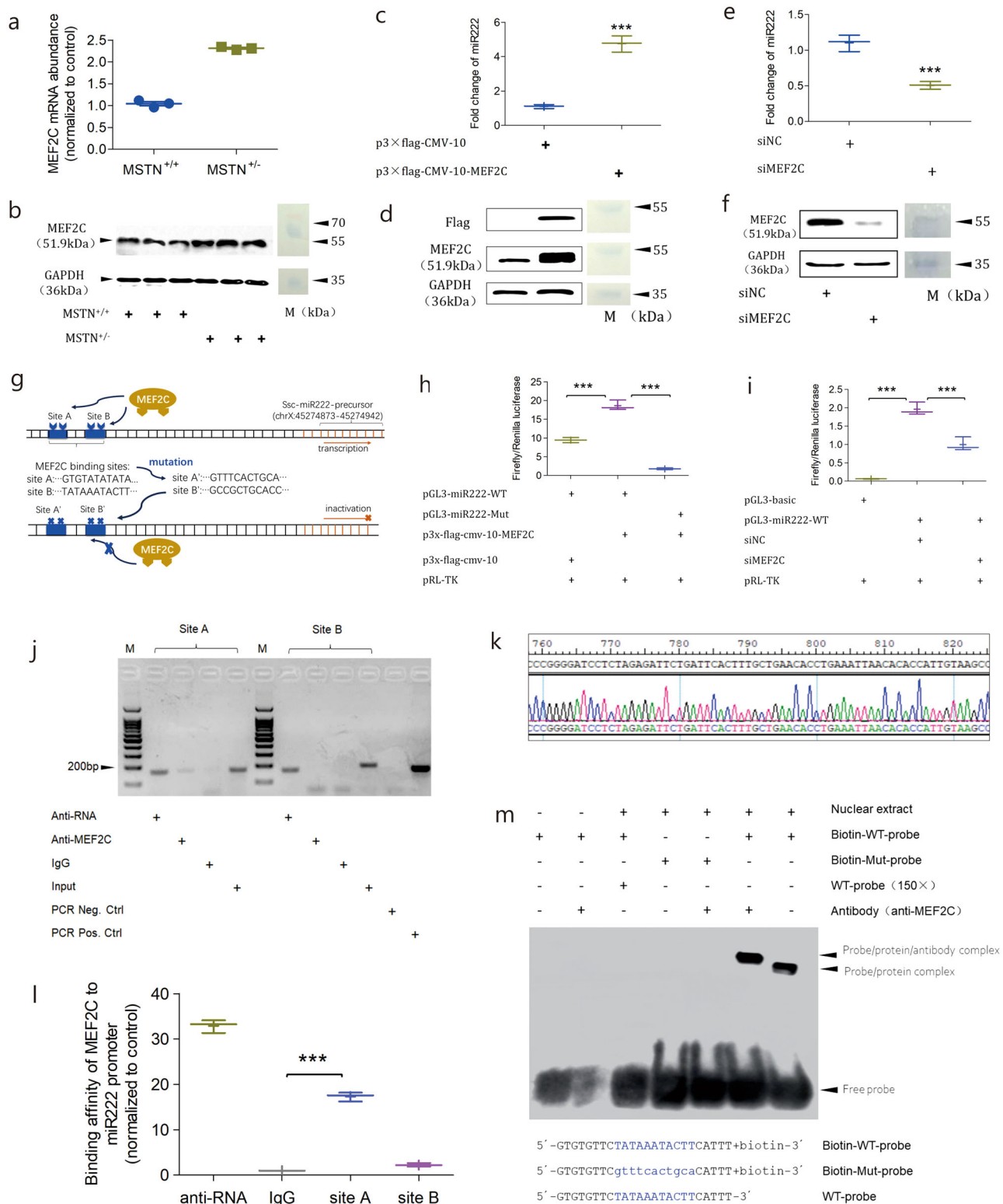

restored MSTN in the MSTN-KO subcutaneous preadipocytes and examined the expressions of these three genes as well as fatty acid composition. In response to the restoration of Flag-tagged MSTN, MEF2C was downregulated at the protein level (Fig. 5a). Concomitantly, MEF2C downregulation reduced miR222 expression and this in turn increased the SCD5 expression, as verified by qPCR and immunoblotting (Fig. 5a, b and Supplementary Fig. 13). Fatty acid composition profiling showed that

the ratios of both C16:1/C16:0 and C18:1/C18:0 rebounded on account of the MSTN restoration. Moreover, on the basis of MSTN restoration, silencing of SCD5 by siRNA was used to examine whether or not the rebound of the fatty acid desaturation was SCD5-dependent. After SCD5 silencing, the ratios of C16:1/ C16:0 and C18:1/C18:0 were reversed again (Fig. 5c and Supplementary Fig. 14). Notably, the expressions of MSTN, MEF2C, and miR222 were not affected by SCD5 silencing, which

**Fig. 4 MEF2C regulates miR222 transcription. a** qPCR of MEF2C mRNA in the subcutaneous fat tissue of MSTN-KO and WT pigs ($n = 3$ biologically independent animals). **b** Western blotting of MEF2C protein in the subcutaneous fat tissue of MSTN-KO and WT pigs ($n = 3$ biologically independent animals). M is the Page Ruler Prestained Protein Ladder 26616 (ThermoFisher Scientific). **c** qPCR of miRNA222 abundance in response to the ectopic expression of MEF2C. $n = 3$ independent experiments. "***" Indicates statistically significant. **d** Western blotting of ectopic MEF2C expression. Flag, flag-tagged MEF2C. M is the Page Ruler Prestained Protein Ladder 26616 (ThermoFisher Scientific). **e** qPCR of miRNA222 abundance in response to MEF2C silencing. Data shown as the mean ± SD ($n = 3$ independent experiments). "***" Indicates statistically significant. **f** Western blotting of MEF2C silencing. M is the Page Ruler Prestained Protein Ladder 26616 (ThermoFisher Scientific). **g** Schematic of the MEF2C-binding sites on miR222 promoter region. "TATAAATACTT", native binding motif. "GCCGCTGCACC", introduced mutant binding motif. **h** Dual luciferase reporter testing of MEF2C upon miR222 promoter. pGL3-miR222-WT and pGL3-miR222-Mut represent wild-type and mutant version of miR222 promoter, respectively. $n = 3$ independent experiments. "***" Indicates statistically significant. **i** Dual luciferase reporter testing of MEF2C silencing upon miR222 transcription. $n = 3$ independent experiments. "***" Indicates statistically significant. **j** ChIP-PCR for the in vivo regulation of MEF2C on miR222 transcription. Sites A and B are the predicted MEF2C-binding sites in the miR222 promoter region. Anti-RNA, positive control antibody against RNA polymerase. Anti-MEF2C, antibody against MEF2C. IgG, negative control antibody. Input, sonicated DNA-protein mixture prior to immuno precipitation. PCR Neg. Ctrl and PCR Pos. Ctrl represent the negative and positive controls to demonstrate the primer specificity. M is 100 bp DNA ladder. **k** Sanger sequencing results for the PCR products from the lane "anti-MEF2C" of site A. **l** ChIP-qPCR validation of MEF2C binding at miR222 promoter. Anti-RNA and IgG are positive and negative controls, respectively. **m** MEF2C binding to the miR222 promoter verified in vitro by EMSA. The sequence and chemical modifications are shown at the bottom. WT probe (×150) was used as a competitor.

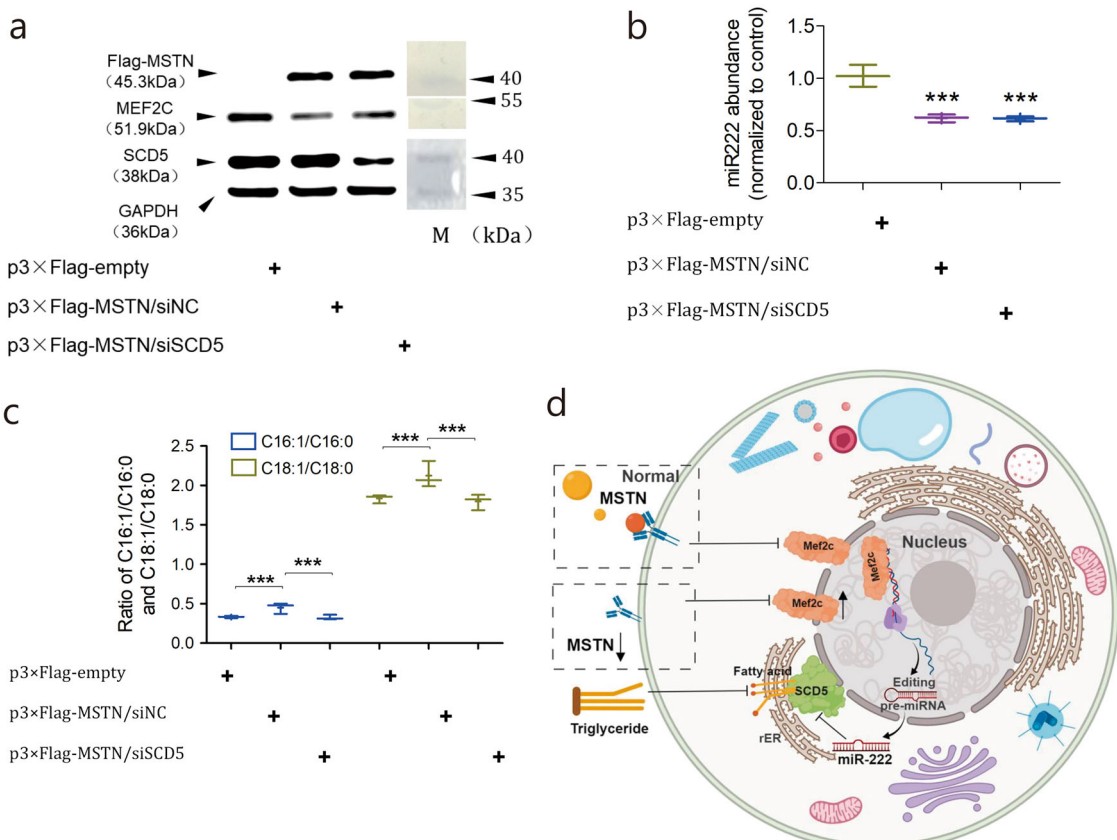

**Fig. 5 MSTN signals through an MEF2C/miR222/SCD5-dependent cascade to affect fatty acid desaturation in pigs. a** Expression of MEF2C and SCD5 in response to the restoration of MSTN detected by Western blotting. MSTN restoration is indicated by the presence of flag-tagged MSTN. GAPDH is the internal control. p3×Flag-MSTN is the MSTN expression plasmid that is tagged by the epitope peptide Flag. siNC and siSCD5 show the treatments with negative control siRNA and siRNA against SCD5, respectively. **b** qPCR detection of miR222 expression in response to the restoration of MSTN. $n = 3$ independent experiments. "***" Indicates statistically significant. **c** Ratios of C16:1/C16:0 and C18:1/C18:0 in response to the restoration of MSTN. $n = 3$ independent experiments. "***" Indicates statistically significant. **d** A model for the MSTN regulation upon fatty acid metabolism through the MEF2C/miR222/SCD5-dependent cascade.

suggests that SCD5 is a downstream regulator in the cascade. Taken together, these data suggest that MSTN-mediated fatty acid desaturation is MEF2C/SCD5/miR222 cascade-dependent (Fig. 5d). In this scenario, the loss of MSTN function results in the upregulation of transcription factor MEF2C in adipose cells. MEF2C increases miR222 expression by binding to its promoter to boost transcription. Upregulated miR222 inhibits SCD5 expression by targeting its 3′UTR. Downregulated SCD5 decreases desaturation of saturated C16:0 and C18:0, which reduces the substrates for TG synthesis. Phenotypically, this leads to less fat deposition, in agreement with the double muscling trait observed in the MSTN-KO pigsl.

## Discussion

Low-efficient HDR in mammals has long been a technical hurdle in exploiting its full potential for precision genetic engineering. However, HDR is a promising route for precise gene editing because it can actualize high-fidelity DNA repair, in contrast to the error-prone NHEJ repair pathway[9]. Researchers thus developed a range of methods to enhance HDR efficiency in mammalian cells. In principle, these improvements fall into four classes, i.e. (i) higher occurrence of DSBs, (ii) inhibition of the NHEJ pathway and/or stimulation of the HDR pathway, (iii) modification of donor DNA template and iv) optimization of selection and enrichment procedures[10]. All these methods enhance HDR efficiency to varying extent.

Here we modified the structure of donor DNA template by adding DsRed as a negative selectable marker, without necessitating the introduction of exogenous small molecules or interfering with the endogenous pathways. A main advantage of DUFAS is that it enables visual discrimination of targeted cell clones from non-targeted cell populations merely via the observation of green and red fluorescence. This design considerably reduces the selection time and avoids the negative selection by a second antibiotics. This is particularly beneficial to gene targeting by HDR in primary somatic cells, which will be further used as donor nuclei for SCNT to produce live cloned animals. In contrast, traditional positive and negative selection will expose the somatic cells to antibiotic selection for a long period, and these cells might be less competent to be reprogrammed for somatic cell cloning[11]. In this study, we applied DUFAS to a total of four sites in immortal and primary porcine cells, and we observed surprisingly high success rates for enrichment of HDR events. We also showed that DUFAS-derived porcine primary targeted cells were competent to be reprogrammed for cloning, resulting in the generation of live, cloned MSTN-KO pigs. We conclude that DUFAS is a robust selection strategy for homolog-directed gene targeting in mammalian cells. We also envision that the combination of DUFAS with other methods to improve HDR efficiency would find wide applications in the precision genome engineering.

Loss-of-function analyses of MSTN in a range of species including rabbit, cattle and sheep, etc, have demonstrated its bidirectional role in increasing the skeletal muscle growth and decreasing the total fat mass[12]. However, the molecular mechanism of how MSTN regulates adipogenesis remains controversial, possibly because its function varies in different cell lineages. A pluripotent cell line C3H10T1/2 can be induced by the glucocorticoid dexamethasone to form adipocytes, and this process can be reinstated by replacing dexamethasone with recombinant MSTN[13]. In comparison, adipogenesis was not induced by the treatment with MSTN in 3T3-L1 cells, which is an adipocyte lineage-committed cell line believed to be further differentiated than C3H10T1/2 cells[4,14,15]. A possible explanation is that MSTN activates SMAD4 to enhance miR124-3p to repress glucocorticoid receptor, resulting in the inhibition of adipogenic differentiation of 3T3-L1 cells[16]. Another possible explanation is that MSTN suppresses 3T3-L1 differentiation by repressing C/EBPα (CCAAT/enhancer-binding protein α) and PPARα (peroxisome-proliferator-activated receptor α), and activating ERK1/2 (extracellular-signal-regulated kinase 1/2)[14]. Regarding its role in terminally differentiated cells, it was reported that MSTN treatment of a mouse hepatocyte cell line promoted fatty acid synthesis and thus significantly increased the TG content, but its underlying action mode was not further addressed[17]. A recent study showed that MSTN regulates the trans-differentiation from myocytes to adipocytes in C2C12 myoblast by affecting Jmjd3 expression through SMAD2/SMAD3 with an impact on the H3K27me3 marker of adipocyte-specific genes[18]. However, the studies summarized above were conducted in mouse cell lines, so it remains unknown whether MSTN employs a different pathway to regulate fat metabolism in pigs.

By intersecting the differentially expressed mRNAs and miR-NAs deeply sequenced in the subcutaneous fat tissues of WT and MSTN-KO pigs, we identified a novel swine-specific molecular pathway, wherein MSTN signals through MEF2C/miR222/SCD5 to affect the fatty acid desaturation and consequently regulate fat deposition. This new pathway features SCD5, which is not present in rodents, and only known to be expressed in humans, cattle, pigs, and birds. SCD5 belongs to the SCD gene family, which are the main rate-limiting enzymes that introduce a double bond at the delta-9 position of saturated palmitic and stearic acyl-CoA for the conversion into palmitoleoyl-CoA and oleoyl-CoA. These two intermediates represent the key substrates for the biosynthesis of complex lipids such as phospholipids and TG[19]. In the pigs, there are two SCD variants SCD1 and SCD5, which were shown to originate from a whole-genome duplication that occurred early in vertebrate evolution[20]. Unlike SCD1, the regulatory circuit, molecular function and physiological effects of SCD5 have not been investigated in pigs to date. Computational analysis showed that porcine SCD5 mRNA sequence has much higher GC content and lacks the N-terminal PEST domain typically present in SCD1. These characteristics indicate that SCD5 might be more stable than its SCD1 counterpart, or is differently regulated[21].

In this study, we validated the molecular function of porcine SCD5, which is capable of catalyzing the conversion of desaturated palmitoleic acid (C16:1) and oleic acid (C18:1) from saturated palmitic acid (C16:0) and stearic acid (C18:0). Porcine SCD5 does not appear to have selective bias on its substrates, as it targets both saturated palmitic (C16:0) and stearic (C18:0) acids without preference. This is particularly promising since these four fatty acids (C16:0, C16:1, C18:0, and C18:1) account for over 80% of the fatty acid composition in pig, and the conversion among them greatly impact the phospho-ligand signaling and whole-body lipid metabolism[22]. Consistent with this notion, in vitro and in vivo experiments demonstrated that TG content was concomitantly affected as a physiological consequence of SCD5 regulation. We speculate that this contribute, at least to some extent, to the reduced fat deposition as observed in the MSTN-KO pigs.

Given the importance of SCD5 in fatty acid composition and fat deposition as revealed in this study, we speculated that its activity should be tightly modulated. We identified miR222 as upstream regulator at posttranscriptional dimension to control SCD5 mRNA translation. In the miRNA profile of human subcutaneous tissue, miR222 was shown to be downregulated during adipogenesis and inversely correlated with the body mass index (BMI), implying its negative regulation upon the adipose tissue generation[23]. In mouse 3T3-L1 preadipocytes, miR222 was also shown to be downregulated during the adipogenesis[24].

Porcine miR222 has been reported to be potentially involved in adipogenesis and fat metabolism. In an inter-breed comparison of miRNA profile, miR222 was shown to be much more abundant in lean-type Landrace pigs than that in Chinese indigenous Lantang pigs, which is consistent with its anti-adipogenic role reported in human and mouse studies[25]. In another study, miR222 was demonstrated to be highly expressed in the adipose tissue of 7-day-old piglets, but it was less expressed in the backfat tissues of 270-day-old adult pigs[26]. Considering that this study used the fatty-type Chinese indigenous Rongchang pigs, it implies that miR222 is inversely correlated with fat mass deposition. Yet, aside from its implication and importance in adipogenesis, miR222's upstream regulators, downstream target genes as well as resulting physiological effect in pigs have not been addressed before. It had

been predicted in silico that the targeting of miR222 to 3′UTR of SCD5 was evolutionarily conserved among the vertebrates[27]. For the first time, here we showed experimentally that miR222 is capable of inhibiting SCD5 expression to affect fatty acid desaturation and consequently the biosynthesis of TG in pigs. Our data imply that the conserved relationship between miR222 and SCD5 is an important element of the regulation of adipose metabolism.

MEF2C has been intensively studied in myofiber type specification as a transcription factor, but recent reports have begun to uncover its role in adipogenesis. In c-Ski deficient mice, MEF2C expression was upregulated by 3.64-fold in white adipose tissues, and inversely correlated with the total fat mass[28]. MEF2C was also shown to be able to activate carnitine palmitoyltransferase 1 (CPT1β) expression in non-muscle cells, and to specifically bind to PPARα in vitro, implying its involvement in the fatty acid metabolism[29]. In pigs, a few studies had validated MEF2C expression in different fat tissues, including intramuscular, subcutaneous, retroperitoneal, and mesenteric adipose tissues[30–32]. In this study, by using in vivo and in vitro assays, we demonstrated that MEF2C is the upstream regulator of miR222 in the adipose tissues of pigs. We speculate that this regulation is an indicator of muscle-fat crosstalk in which a myokine can execute its function in adipogenic environments. Although we have shown that miR222 upregulation by MSTN is MEF2C-dependent, the mechanism behind MSTN and MEF2C interaction requires further clarification.

Mounting evidence in animals has revealed the importance of MSTN in lipid metabolism. Transcriptomic analyses in MSTN knockout mouse, goat, and cattle showed that MSTN can affect a number of genes directly involved in fatty acid metabolism like SCD and ELOVL[33–35], but the inherent link between MSTN and adipogenic genes has not been established. In this study, we took advantage of RNA-seq to identify the potential miRNA-mRNA regulatory pairs in the fat tissues of MSTN-KO and WT pigs. By several lines of in vitro and in vivo assays, we validated the post-transcriptional interaction between miR222 and SCD5, as well as the transcriptional regulation of MEF2C upon miR222. Our study provides insights into how MSTN signals through the MEF2C/miR222/SCD5 cascade to regulate the fatty acid desaturation and further affect the TG biosynthesis and fat deposition in pigs. To the best of our knowledge, this is the first report that identified a signaling pathway through which MSTN impacts fatty acid metabolism.

In summary, given the same origin of myocytes and adipocytes deriving from mesodermal precursor cells, we conclude that MSTN, as a master regulator of double muscling trait in animals, has a functional partition to coordinate both adipogenesis and myogenesis. This functional partitioning will be beneficial for pig breeding for high leanness and high-IMF content pork.

## Methods

**Ethics statement and animal samples**. All experimental procedures involving animals were reviewed, approved, and supervised by the Animal Care Committee of the Institute of Animal Science and Veterinary Medicine, Hubei Academy of Agricultural Sciences, China. All the WT and MSTN-KO pigs (male, $n = 3$, respectively) were raised with same diet and conditions in the pig farm affiliated with the Institute of Animal Science and Veterinary Medicine (Jiangxia District, Wuhan, China). Samples of the subcutaneous fat tissues were collected, snap frozen in liquid nitrogen and preserved at $-80\,°C$ until RNA or protein extraction.

**Growth performance and slaughter trial**. Growth performance was tested in the Feed Intake Recording Equipment system (Orborne Industries Inc., Orborne, KS, USA). Slaughter trials, including carcass character and meat quality examinations, were performed in the Breeding Swine Quality Supervision, Inspecting and Testing Center in Wuhan (Ministry of Agriculture, China), when the pigs were at 7 months of age and ~70 kg of weight. Blood metrics were examined as previously described[36].

**Plasmids, strains, and cell lines**. CRISPR/Cas9/gRNA expression plasmid px330 (Cat. No. 42330) was obtained from Addgene. The dual luciferase plasmid pGL3-promoter, pGL3-basic, pRL-TK, dual-fluorescence targeting plasmid backbone pNEO-EGFP-RFP, DH5α E. coli strain and porcine cell line PK15 were preserved by the Key Laboratory of Animal Embryo Engineering and Molecular Breeding of Hubei Province (affiliated to Institute of Animal Science and Veterinary Medicine, Hubei Academy of Agricultural Sciences). The TA cloning plasmid was purchased from Takara (Dalian, China). Competent E. coli DH5α cells were prepared according to the standard protocols. The Pig primary fetal fibroblast cells were prepared as previously described[36]. Porcine subcutaneous preadipocytes were isolated aseptically from backfat tissue of five-day-old Meishan piglets killed via anesthesia. Briefly, ~1 g of the muscle samples were collected aseptically and cut into 1 mm³ cubes with ophthalmic scissors in a sterile culture dish. This was followed by three rounds of washing with DPBS buffer in a 50 ml centrifuge tube, and digestion with 0.2 U μl⁻¹ Type II collagenase for 2 h at 37 °C in a water bath. After full digestion, the cell suspension was filtered in batches through a 74 μm cell sieve mesh and stored in new centrifuge tubes, removing tissue debris. After centrifugation at $2000 \times g$ for 5 min, the cell pellet was resuspended in DMEM/F12 medium following with filtration of cell suspension through a 37 μm cell sieve mesh. After re-centrifugation, cell pellets were again resuspended with DMEM/F12 medium. Cells were then evenly inoculated in a sterilized culture flask supplied with culture medium and cultured at 37 °C under 5% $CO_2$ in a humidified incubator. It took ~2 h for subcutaneous preadipocytes to reach the adherent culture stage. The adherent cells were then washed twice with DMEM/F12 medium and then supplied with new culture medium. The culture medium was refreshed every 3 days depending on confluency.

To construct the donor templates for homologous recombination, homologous arms targeting T1, T2, and T3 sites in MSTN exons were cloned into pNEO-EGFP-RFP plasmid. Flag-tagged porcine MEF2C, MSTN and SCD5 cDNAs were cloned into the p3×Flag-CMV-10 backbone to make the p3×Flag-CMV-10-MEF2C/MSTN/SCD5 expression plasmid, respectively. The porcine miR222 promoter was cloned into the pGL3-basic to make the reporter plasmid pGL3-basic-miR222-WT. MEF2C-binding site was mutated by a Mut Express MultiS Fast Mutagenesis Kit V2 (C215-01/02, Vazyme, Nanjing, China) to make the mutant reporter plasmid pGL3-basic-miR222-Mut. The 3′UTR of porcine SCD5 was cloned into the pGL3-promoter plasmid to make the reporter plasmid pGL3-SCD5-UTR-WT. The binding site of miR222 on the 3′UTR of SCD5 was mutated as described above to make the mutant reporter plasmid pGL3-SCD5-UTR-Mut.

### Transcriptome sequencing

*Library construction and sequencing*. Total RNAs were extracted according to the manufacturer's instructions of Trizol reagent (Invitrogen). RNA quality was examined using a Labchip GX Touch HT Nucleic Acid Analyzer (PerkinElmer, USA). mRNA was purified using oligo(dT), fragmented and primed with random hexamer primers. Double-stranded cDNA was synthesized and purified with AMPure XP beads. The quality-controlled cDNA was used for sequencing with PE150 on Illumina. Small RNA libraries were generated using a NEBNext® Multiplex Small RNA Library Prep Set for Illumina® (NEB, USA) following manufacturer's recommendations and index codes were added to attribute sequences to each sample. Briefly, libraries were prepared by ligating different adaptors to the total RNA followed by reverse transcription, PCR amplification and size selection using 6% polyacrylamide gels. Library quality was assessed on the Agilent Bioanalyzer 2100 system and sequenced on the Illumina platform.

*Reads processing and mapping*. Raw data (raw reads) of fastq format were firstly processed through in-house perl scripts. In this step, clean data (clean reads) were obtained by removing reads containing adapter, reads containing ploy-N and low-quality reads from raw data. At the same time, Q20, Q30 and GC content of the clean data were calculated. All the downstream analyses were based on the clean data. The reference pig genome and gene model annotation files were downloaded from https://genome.ucsc.edu/. We selected Hisat2 (v2-2.0.0-beta) as the mapping tool. The number of perfect clean tags for each gene was calculated and then normalized to reads per kilobase of exon model per million mapped reads (RPKM). Finally, all of the related genes (referred to KEGG pathways database) with RPKM > 0 were chosen and compared by $\log_2$ (RPKM). The clean reads from small RNA libraries were made non-redundant using miRDeep2 software (v2.0.0.8), and the collapsed reads were compared with the reference genome to identify small RNA species. The known miRNA sequences of species were obtained from miRBase (http://www.mirbase.org/). The information of known miRNA expression levels in samples and the prediction of novel miRNAs were obtained using miRDeep2.

*DE mRNA and miRNA identification*. Differential expression analysis of the MSTN-KO and WT pigs ($n = 3$, respectively) was performed using the DESeq R package (v1.10.1). DESeq provides statistical routines for determining differential expression in digital gene expression data using a model based on the negative binomial distribution. The resulting $p$-values were adjusted using the Benjamini and Hochberg's approach for controlling the false discovery rate. mRNAs and miRNAs with an adjusted $p$-value < 0.05 and abs($\log_2$FoldChange) >1 found by DESeq were categorised as differentially expressed.

*miRNA target and TF prediction*. The miRNA target prediction was carried out by miRanda (http://www.microrna.org/microrna/getMirnaForm). The prediction metrics were as follows: energy threshold set at −20 kcal/mol with default = 1; score threshold set at default of 140; output parameter was set to single out top 50 target genes with the highest score. TF prediction was performed by the web server at http://alggen.lsi.upc.es/cgi-bin/promo_v3/promo/promoinit.cgi?dirDB=TF_8.3, in which "transcription factors and sites of all species" was chosen and maximum matrix dissimilarity rate <15% was applied.

*GO and KEGG enrichment analysis of differentially expressed genes*. GO and KEGG enrichment analyses were performed using the clusterProfiler package (v3.6.0) from BioConductor. Significantly enriched GO terms and pathways were selected according to the *p*-value < 0.05. ClusterProfiler R package was used for enrichment map drawing. We used Pathview R package (v1.18.2) to test the statistical enrichment of differentially expressed genes in KEGG pathways (http://www.genome.jp/).

**DNA and RNA manipulations**. All DNA oligos used in this study were chemically synthesized by Sangon Biotech (Shanghai, China) and listed in Supplementary Table 6. Small interfering RNAs (siRNA) and miRNAs were designed and chemically synthesized by Ribobio (Guangzhou, China). The targeting sequence of siRNA against porcine SCD5 was 5′-TCCAGAGAAAGTACTATAAGATC-3′. The targeting sequence of siRNA against MEF2C was 5′-GATGCCATCAGTGAATCAA-3′. PureLink® HiPure Plasmid Filter Purification Kits were used for midi and maxi preparation of all plasmid DNAs (Invitrogen). The genomic DNAs were extracted using PureLink Genomic DNA Kit (Invitrogen). Total RNA extraction was carried out using Trizol reagent (Invitrogen). The concentration and purity of genomic DNA and total RNA were determined by a NanoDrop spectrophotometer (ThermoFisher). Integrity of genomic DNA and total RNA was examined by agarose gel electrophoresis. Southern blotting was performed as previously described[36]. Small RNAs were isolated with a Purelink™ miRNA Isolation Kit (K157001, Invitrogen) according to the manual. For the Northern blotting, 10 μg small RNAs were denatured for 5 min at 72 °C to disrupt the secondary structure, and placed on ice for 10 min to keep single-stranded. Small RNA were then electrophoresed for 1 h on a 7 M Urea/15% acrylamide gel at 200 V, and transferred to a Hybond NC membrane (Amersham). The membrane was cross-linked twice at 254 nm of UV light at 0.35 J cm$^{-2}$ using a HL-2000 hybrilinker UV crosslinker (UVP). After prehybridization for 1 h at 40 °C, the membrane was hybridized with DIG-miRNA222 probe at 40 °C overnight. The membrane was washed with 2x SSC, 0.1% SDS (5 min at room temperature, twice), 0.5x SSC, and 0.1% SDS (15 min at 42 °C, twice) and finally colorized with the DIG high prime DNA labeling and detection starter kit (Roche).

**End-point PCR and qPCR**. cDNA synthesis was carried out using a PrimeScript RT reagent kit (Takara). LA Taq (Takara) DNA polymerase was used in the end-point PCR testing. A typical LA Taq PCR reaction mixture contained 5 μl 10×LA PCR buffer (Mg$^{++}$ plus), 8 μl dNTP mixture (2.5 mM each), LA Taq polymerase 0.5 μl (5 units/μl), forward and reverse primer 1 μl (20 μM, respectively), template 0.5 μg and PCR-grade water added to 50 μl in total. Cycling condition were 94 °C 2 min, 30 cycles of 94 °C 30 s, 55 °C 30 s and 72 °C 1 kb min$^{-1}$, followed by a final extension of 5 min at 72 °C. One fifth volume of the PCR product was fractionized by either 1% or 2% agarose gel and photographed by the ChemDOC™ XRS$^+$ (Bio-Rad Laboratories, Inc., Hercules, CA,USA). qPCR was performed on real-time thermocycler (Roche lightcycle 96) with real-time PCR TB Green Premix Ex Taq II (Takara). According to the manufacturer's instructions, the qPCR reactions were conducted in 20 μl of reaction buffer containing TB Green Premix Ex Taq polymerase II, 0.8 μl of 10 M forward and reverse primers, 2 μl of cDNA and 6.4 μl sterilized water. The qPCR reaction system was performed with the following steps: 2 min at 94 °C followed by 45 cycles of 30 s at 94 °C, 30 sec at 60 °C, and 30 s at 72 °C, followed by a melting cycle from 55 °C to 95 °C to assess the amplification specificity. The RNA expression levels were normalized to the level of GAPDH expression while U6 was used as housekeeping gene for miRNA. Relative expressions were calculated using the comparative $2^{-\Delta\Delta CT}$ method. Paired *t*-test (significant *p*-value < 0.05, two-tailed) was employed to assess the significance of difference.

**Cell culture and transfection**. PK15 cell line was cultured and maintained in Dulbecco's Modified Eagle's Medium (DMEM, Gibco) supplemented with 10% (v/v) fetal bovine serum (FBS, Hyclone) and 10 μg/ml of penicillin-streptomycin solution (Sigma-Aldrich Corp.) at 38.5 °C in a humidified atmosphere of 5% $CO_2$ and 95% air. The culture medium was refreshed every other day to ensure optimal proliferation. Electroporation was conducted as previously described[36]. Liposome transfection for both siRNA and miRNA were performed using Lipofectamine 3000 (Invitrogen) as described in the manufacturer's protocol. Dual-luciferase assays were performed according to manufacturer's protocol (Promega) and detected in a GloMax Multi luminometer (Promega). Each group included three technical repeats. Pig SCNT was carried out mainly as previously described[36].

**ChIP**. The ChIP assays were performed using the Chromatin IP Kit (17-371, Millipore) according to the manufacturer's protocol. Intramuscular preadipocytes were cultured in a 150 mm culture dish containing 20 mL growth medium. When they reached 80% cell density, cells for the treatment group were transfected with the p3×-flag-cmv-10-MEF2C plasmid, followed with 72 h of cell culturing. In all, $1 \times 10^7$ cells in 150 mm dishes were cross-linked with 1% formaldehyde in the culture medium for 10 min at room temperature. The surplus formaldehyde was quenched by the addition of one-tenth volume of 1.25 M glycine (pH 7.0) for 5 min. The cells were washed with PBS, dislodged by scraping, collected by centrifugation at 800 × *g* for 5 min at 4 °C, resuspended in the cell lysis buffer (10 mM Tris HCl, pH 7.5, 10 mM NaCl, 0.5% NP-40), and incubated on ice for 15 min. DNA was sheared using a sonication system (VC 130PB, SONICSC & MATERI-LAS INC, USA) with 130 Watts, 30% power, 10 impacts, 5 s per impact. Aliquots of sheared chromatin were immunoprecipitated using ChIP Blocked Protein G Agarose (Catalog No. 16-201D, Merck) and 5 μg of the above-mentioned MEF2C antibodies. Immunoprecipitation with Anti-RNA Polymerase II (Catalog No. 05-623B, Merck) and normal mouse IgG (Catalog No. 12-371B, Merck) were used as positive control and negative controls, respectively. After immunoprecipitation, crosslinks were reversed by heating to 65 °C, and immunoprecipitated DNA was purified using spin columns (Cat. No. 54D107, NINGBO SCIENTZ BIO-TECHNOLOGY, China). The eluted free DNA was purified using a DNA purification kit (Cat No. D3396-01, OMEGA). qPCR analysis of the ChIP and genomic input DNAs was performed using SYBR® Green II (Cat. No. RR820, TAKARA) according to the supplier's protocol.

**EMSA**. Nuclear extracts were prepared from intramuscular preadipocytes transfected with the MEF2C plasmid using a commercially available kit from ThermoFisher (Cat. No. 78833). Synthetic oligonucleotides 3′-labeled with biotin (Sangon Biotech) were annealed to generate double-stranded oligonucleotides as probes. EMSA was performed according to the instructions provided with the LightShift Chemiluminescent EMSA kit (Cat. No. 20148, ThermoFisher). Sample preparation included recommended volumes of binding buffer, poly (dI·dC), 50% glycerol, 1% NP-40, and 100 mM MgCl$_2$. Then 4 μl of 5x loading buffer was added, and the sample was electrophoresed through a 6% non-denaturing polyacrylamide gel in 0.5x TBE at 4 °C. For the competition experiments, unlabelled wild-type oligonucleotide was added in a 150 pmol excess prior to the addition of the biotinylated probe. To identify the transcription factor present in the DNA protein complex using a supershift assay, the nuclear extracts were incubated in the binding buffer for 60 min at 4 °C in the presence of MEF2C antibody (Cat. No. EPR19089-202, Abcam; dilution 1:1,000) prior to the addition of the biotinylated probe. Samples were run on 6% DNA retarding gels (Cat. No. EC6365BOX, Invitrogen) and transferred onto 0.45 μm nylon membranes (Cat. No. 77016, Biodyne) for detection on a GeneGnome XRQ NPC CCD camera (Serial Number SYGN0/04186).

**Western blotting**. Total cellular protein extracts were collected in the RIPA Lysis and Extraction Buffer (ThermoFisher). Protease inhibitor (PMSF) at concentration of 1% was added into the RIPA buffer to avoid proteolysis. For each sample, 50 μl protein lysate collected in a 1.5 ml centrifuge tube was treated with ice-bath for 30 min, followed by centrifugation at 10,000 × *g* at 4 °C for 5 min. Protein supernatant in the tube was transferred into a new tube and stored at 4 °C for further protein quantification by BCA Protein Assay Kit (P0006, Beyotime Institute of Biotechnology). In all, 20 μg of protein extract for each sample was subjected to 12% sodium dodecyl sulfate-polyacrylamide gel electrophoresis (SDS-PAGE Gel Preparation Kit, Beyotime) and transferred to Polyvinylidene difluoride (PVDF) membrane (Bio-rad). The membranes were blocked in TBST (150 mM NaCl, 20 mM Tris-HCl at pH 8.0, 0.05% Tween 20) blocking buffer with 5% non-fat dry milk powder at room temperature for 1 h. After three washes with TBST buffer, the membranes were incubated with primary antibodies, overnight at 4 °C. After three washes in TBST buffer, blots were incubated with secondary antibodies. Protein bands were then visualized by ECL detection reagent (BeyoECL Moon, Beyotime). Band identification and quantification were conducted using a ChemiDoc™ XRS$^+$ System and Image Lab Software (BioRad). The primary antibodies used for Western blotting were rabbit monoclonal antibody against the mouse MEF2C (ab231859, Abcam; dilution 1:1,000), rabbit polyclonal antibody against the mouse MSTN (TA343358, Origene; dilution 1:1,000), mouse monoclonal antibody against the human GAPDH (60004-1-Ig, Proteintech, Wuhan, China; dilution 1:2,000), rabbit polyclonal antibody against the human SCD5 (ab130958, Abcam; dilution 1:1,000), mouse monoclonal antibody against the human β-tubulin (66240-1-Ig, Proteintech, Wuhan, China; dilution 1:1000) and mouse monoclonal antibody against the Flag (F1804, Sigma-Aldrich; dilution 1:2,000). HRP-conjugated secondary antibodies were anti-mouse and anti-rabbit IgG (7076 and 7074, respectively, Cell Signaling Technology; dilution 1:2,000).

**TG content measuring**. Total TG content was measured using a tissue triglyceride assay kit (E1003-2 from Applygen Technologies Co., Ltd., Beijing, China). One hundred microliter lysis buffer was added to homogenized cell culture in one well of a 24-well plate. The homogenate was incubated in water bath at 70 °C

for 10 min, centrifuged at $2,000 \times g$ for 5 min. The lysate pellets were used for protein quantification of samples (BCA Protein Assay Kit, Beyotime, China). For a 96-well plate, 10 µl of supernatant was subjected into one well with 190 µl of working solution (triglyceride assay kit), triplicate for each sample. After incubating at 37 °C for 10 min, a microplate reader (VictorTM X5; PerkinElmer) was used to detect absorbance values at 550 nm. The TG content was normalized to the protein concentration.

**Histology.** The preadipocytes were stained with Oil Red O. Briefly, the cells were washed gently with pre-warmed DPBS twice and then fixed with 4% paraformaldehyde for 45 min. One milliter of the ready-to-use Oil Red O solution was slowly added to cells for staining for 30 min. The colorimetric development of the lipid droplets was observed under the microscope and photographed. Five randomly selected fields of the Oil Red O-stained preadipocytes were quantified using the Image J software (NIH, USA).

**Fatty acid profiling.** Methyl nonadecanoate (250 mg) was obtained from Aladdin Industrial Corporation (America). Methyl docosapentaenoate (10 mg) was obtained from Nu-chekprep Corporation (America). Supelco 37 component fame mix (CRM47885) was obtained from Sigma-Aldrich Corporation (America). Methonal and heptane were obtained from Thermo Fisher Scientific Corporation (America), both of which were chromatographic grade. Boron trifluoride-methanol solution (14%) was obtained from Sigma-Aldrich Corporation (America). Sodium hydroxide and Sodium chloride were obtained from Sinopharm Chemical Reagent Corporation (China), both of which were analytical grade. All reagents used were analytical reagent grade or better.

Methyl nonadecanoate was used as an internal standard. The stock solutions with concentration of $1000\,\mathrm{mg\,L^{-1}}$ were prepared by dissolving methyl nonadecanoate and methyl docosapentaenoate into heptane, and then diluted 10-fold to $100\,\mathrm{mg\,L^{-1}}$ standard solutions with heptane, respectively. Two hundred micro liter methyl nonadecanoate and methyl docosapentaenoate standard solutions ($100\,\mathrm{mg\,L^{-1}}$), 100 µl Supelco 37 component Fame mix and 500 µl heptane were mixed in a 2 ml glass vial to obtain the standard solution.

Harvested cells were pulverized under liquid nitrogen for 15 min. For saponification, 100 mg of pulverized cells was put into a 50 ml conical flask and was dissolved in 2 ml of 2% sodium hydroxide methanol solution with the addition of 60 µl methyl nonadecanoate stock solution ($1,000\,\mathrm{mg\,L^{-1}}$) as internal standards. A conical flask was incubated at 70 °C in a water bath for 30 min for heat reflux. Then 5 ml of 14% boron trifluoride-methanol solution was added to the flask from the top of the condenser pipe and water bath at 70°C for 30 min. The heat reflux unit was then taken out from the water bath and naturally cooled down to ambient temperature. Then 2.94 ml heptane was added to the flask and water bath at 70 °C for 5 min. After naturally cooling to ambient temperature, 20 ml of saturated sodium chloride solution was then added into the flask, and the flask shook thoroughly. A sufficient amount of saturated sodium chloride solution was poured into a 50 ml conical flask to reach the scale line. Once the standing and layering processes completely finished, the upper organic phase was taken as fatty acid methyl esters (FAMES) for subsequent analyses.

The processed samples were analyzed by a GC-MS system (GCMS-TQ8040, Shimadzu Corporation, Japan) and the mass spectrometer detector was equipped with electron impact ionization mode (EI, 70ev). An Agilent J&W HP-88 capillary column (100 m × 0.25 mm × 0.20 µm) and a selected ion monitoring mode (SIM) were used to separate and detect different possible FAMES. The splitless injector port was maintained at 220 °C and the volume of the injected sample was 1.0 µl. High purity of helium (≥99.999%) was used as carrier gas ($1\,\mathrm{mL\,min^{-1}}$). The oven temperature was raised from 60 °C (held for 1 min) to 140 °C at $40\,^{\circ}\mathrm{C\,min^{-1}}$, and then increased at 4°C/min to 240 °C (held for 15 min), for a total run time of 43 min. The interface temperature of MS was maintained at 250 °C and ion source temperature was maintained at 230 °C. All samples were analyzed in triplicate. The above mixed standard solution was used as analyte standard. The response factors of the FAMES peak areas to the methyl nonadecanoate peak area against the analyte standard concentration were calculated, and then the quantitative analysis of FAMES was performed using the inner standard calibration method. The individual peaks were identified by comparison with retention times and characteristic fragment ions of standard chromatogram produced by the analyte standard. Signal collection and data analyses were conducted using GCMS solution software (v4.30) in the Beijing Center for Physical & Chemical Analysis (Beijing, China).

**Statistics and reproducibility.** Band intensities were quantified by ImageJ (NIH, USA). Statistical comparisons of the growth performance, carcass characters, meat quality, and blood index were carried out by Student's $t$ test in Graphpad Prism 4.0 analyzer (Graphpad Software, La Jolla, CA, USA) and $p < 0.05$ was considered as statistically significant. All the values were shown as mean ± SEM (standard error of the mean) of independent biological replicates. Phenotyping, genotyping and RNA-seq were performed in triplicates ($n = 3$ biologically independent samples). Western blots, Southern blots, Northern blots, microscope-based data, agarose gel images, histology, fatty acid profiling are representative of multiple replicates showing similar results.

**Plotting tools.** All artworks were created using CorelDRAW Graphics (Corel Corporation, Ottawa, Canada).

## Data availability

The deep sequencing data have been deposited in SRA database under the accession numbers: SAMN14089950, SAMN14089951, SAMN14089952, SAMN14089953, SAMN14089954, SAMN14089955. The targeting plasmid backbone and expression plasmids for porcine MSTN, MEF2C and SCD5 have been deposited in Addgene with the accession numbers: 157952, 157953, 157954, and 157955. All other data are available from the corresponding author (Y.B.) upon reasonable requests. Source data underlying plots shown in figures are provided in Supplementary Data 1. Images of full gels and blots are shown in Supplementary Information.

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

## Acknowledgements

We are grateful to Wuhan Jinying Animal Science Company Ltd. for assistance in animal raising. We thank Dr. Youxiang Zhou (Hubei Academy of Agricultural Sciences, Wuhan, China) for his technical assistance in the fatty acid profiling. We thank Dr. Adrian Molenaar (AgResearch Ltd., New Zealand) for his constructive discussion about the paper. This study is supported by the National Natural Science Foundation of China (31772577) and the China Major Program of Genetically Modified Organism New Variety Cultivation (2016ZX08006001-005). It is also supported by the Innovation Center for Agricultural Sciences and Technologies of Hubei Province (2020-620-000-001-20), Outstanding Young Scientist of Natural Science Foundation of Hubei Province (2018CFA043), and Leading Scientist Cultivation Program of Hubei Academy of Agricultural Sciences (L2018015).

## Author contributions

Y.J. and Y.B. conceptualized and designed the study. H.R., W.X., X.Q., G.C., H.C. C.C., J.D., and Z.Z. performed the molecular and cellular biology experiments and analyzed data. Z.H., W.H., H.X., L.Z., and X.Z. performed the porcine embryology experiments, phenotypic testing and analyzed data. X.Q. and X.L. performed the bioinformatic analysis. C.Q. performed the fatty acid profiling. H.R., W.X., H.C. and Y.B. wrote the paper. All authors read and approved the final paper.

## Competing interests

The authors declare no competing interests.
