## [Peer Review File · Communications Biology]

Reviewers' Comments:

Reviewer #1:

Remarks to the Author:

1. In the 4.3 DE mRNA and miRNA identification of the method section. 3. The author need to introduce the details of the DESeq command used, because each group in your experiment has only two biological repetitions. In addition, you are also very inaccurate in describing the selection of differential expression. not only does you not provide fold change, but $P < 0.05$ is not the judgment standard for differential genes, but it is significant. Please reply and modify it accurately here.
2. In the 4.4 miRNA target and TF prediction of the method section.
3. The author should provide detailed filtering conditions for miRNA and transcription factors, such as matching rate and P value.
4. In the 4.5 GO and KEGG enrichment analysis of differentially expressed genes of the method section. Please provide the enrichment conditions and enrichment map drawing software package of KEGG here. The author said they used the software, and which software is it and can you provide the readers with detailed information?
5. In the 14. Plotting tools and statistics of the method section. The exact name of the statistical tool used by the author is "GraphPad Prism 4.0." Please correct it. In addition, which test method is used for PCR results? Please include it.
6. In the Supplementary Fig. 5-8 of supplementary material section, there are single and plural syntax errors in the title of the picture, please check carefully.
7. I seldom see such a writing style as miRNA: mRNA network. Can miRNA-mRNA network be ok?
8. The author's annotations on the figures are very confusing. Can he describe the content of the figure instead of explaining it? The contents of these explanations must be put into the results, such as "fig5d..... Phenotypically, this lead to less fat deposition, in agreement with the double muscling trait observed in the MSTN KO pigs..". For another example, the annotation of fig. 2-b-c are combined, while the volcano map is in singular form. In addition, can fig2-f add a line to help explain which column is the control group?
9. Figures (fig2-b,c,d,e) of high-throughput sequencing are not clear enough, please update them.
10. In the supplement document, the title should be placed below the picture, not above it.
11. The author must first ask people from professional fields with excellent English writing ability to help rewrite the complete manuscript, and then ask professional English polishing service agencies to do the second round of proofreading. The current English writing is completely below the published standard.

Reviewer #2:

Remarks to the Author:

Ren et. al. sought to better understand the link between MSTN disruption in livestock (pigs in this case) and reduction in intramuscular fat content in MSTN mutant animals. To do this, they first developed a clever HDR template/system (DUFAS) that incorporates positive/negative selection using a combination of drug selection and fluorescence. While effective for producing the founder animals in this study, details around the method and data were limited and the overall utility of such a method is unclear. However, the primary novelty of the paper is not related to the engineering method, but the downstream analysis of the MSTN mutant pigs. The authors found that their pigs phenocopied the numerous other reports of MSTN deficiency in livestock. Having demonstrated this, they leveraged RNAseq of subcutaneous fat to identify differentially expressed mRNAs and miRNAs. Using a series of ontology tests and miRNA:mRNA pairing studies, they homed in on 5 mRNA nodes with differential expression miRNA:mRNA pairs. From the list of 5, they chose SDC5 to investigate further, though it was not clear why if this was serendipitous, due to functional ontology, or literature review since the SDC5 node was much less prominent in the miRNA:mRNA network. Regardless, the authors ran a series of in vitro and biochemical studies

that clearly linked miR222 to SCD5 regulation and further identified the motif in the 3' UTR of SCD5. To link miR222 increase to MSTN reduction, they evaluated differentially expressed transcription factors against the TF binding sites in the miR222 promoter. This analysis revealed that the MEF2C, overexpressed in MSTN mutant pigs, was at least partially responsible for miR222 overexpression. The link was proven through in vitro expression of MEF2C shown to increase miR222, and biochemical evidence that MEF2C binds to the miR222 promoter. While the link between MSTN and MEF2C is not clear, the authors show that restoration of MSTN reduces MEF2C and in turn increases SCD5 activity. Further in vitro assays with adipocytes demonstrate an alteration in fatty acid composition as a result of this pathway, consistent with the observations of samples collected from the pigs. Taken together, the authors have revealed a putative separation in the pleiotropic effects of MSTN knockout, where alteration of a second gene, SCD5, could have a role in restoration of intramuscular fat in MSTN KO animals. However, while this link is established, upstream role in adipocyte differentiation and quantity could only be speculated based on studies in other systems, so additional experimentation is required to determine the magnitude of effect when intervening at different levels of this pathway. Overall, I think the work to discover the MSTN/MEF2C/miR222/SCD5 link is of high quality and supported in numerous ways. As mentioned, the magnitude of effect in vivo will be very interesting. The rationale for DUFAS and its utility are somewhat in question by this reviewer, this data could be minimized in the overall story as the data is not their to support their claims and the resulting story is much more interesting.

1. Methods and data for DUFAS were significantly lacking for both PK15 and fibroblasts.
 - a. Was Neo selection applied?
 - b. What was the distribution of Neo + clones with one or both fluorescent markers?
 - c. What was the prevalence of perfect 5' and 3' integration in different classes of clones.
 - d. For HDR positive clones in Sup Table 1, how were they validated? Were they confirmed at the 5', 3' junctions, or both?
 - e. In Sup Fig 2 and Figure 1, are the "5 Clones" shown in the gel the same 5 clones from left to right for both 5' and 3' junctions?
2. Several labs have reported high efficiency HDR using oligonucleotide templates that leave no remnants at the target site. When and why would DUFAS be applied if that is the case? Some examples (Wang, 2016 NAR; Tan, 2013 PNAS)
3. Figure 2 is nearly impossible to read- I presume this is due to lower resolution of the reviewer PDF, but consider increasing font. I was not adequately able to assess 2d and 2e since they could not be read.
4. What factors lead to selection of SCD5? There were 4 other nodes more prominent on the network, so please clarify the rationale.
5. Under subheading 5, you made the following statement "Concomitantly, MEF2C downregulation reduced miR222 expression and this further lowered SCD5 expression, as verified by qPCR and immunoblotting (Fig. 5a, b)." Should this statement state that SCD5 expression was increased?

	Comments of reviewer 1	Reply
1	In the 4.3 DE mRNA and miRNA identification of the method section. 3. The author need to introduce the details of the DESeq command used, because each group in your experiment has only two biological repetitions. In addition, you are also very inaccurate in describing the selection of differential expression. not only does you not provide fold change, but P<0.05 is not the judgment standard for differential genes, but it is significant. Please reply and modify it accurately here.	More accurate description was given to this section. Please refer to line 548-554.
2	In the 4.4 miRNA target and TF prediction of the method section. The author should provide detailed filtering conditions for miRNA and transcription factors, such as matching rate and P value.	More accurate description was given to this section. Please refer to line 556-563.
3	In the 4.5 GO and KEGG enrichment analysis of differentially expressed genes of the method section. Please provide the enrichment conditions and enrichment map drawing software package of KEGG here. The author said they used the software, and which software is it and can you provide the readers with detailed information?	The missing information has been added to this section. Please refer to line 565-570.
4	In the 14. Plotting tools and statistics of the method section. The exact name of the statistical tool used by the author is "GraphPad Prism 4.0." Please correct it. In addition, which test method is used for PCR results? Please include it.	The name of the tool had been corrected as per the instruction. $2^{-\Delta\Delta CT}$ method was used for the PCR results, as indicated in the subhead 6 of the method section. Please refer to line 614 and 771.
5	In the Supplementary Fig. 5-8 of supplementary material section, there are single and plural syntax errors in the title of the picture, please check carefully.	These errors had been corrected. Please refer to the supplementary file.
6	I seldom see such a writing style as miRNA:mRNA network. Can miRNA-mRNA network be ok?	All the "miRNA:mRNA" has been modified to "miRNA-mRNA" throughout the text.
7	The author's annotations on the figures are very confusing. Can he describe the content of	The legends of all the five figures had been revised.

	the figure instead of explaining it? The contents of these explanations must be put into the results, such as “fig5d..... Phenotypically, this lead to less fat deposition, in agreement with the double muscling trait observed in the MSTN KO pigs..”. For another example, the annotation of fig. 2-b-c are combined, while the volcano map is in singular form. In addition, can fig2-f add a line to help explain which column is the control group?	Description of the contents were shown in the legends, while the explanations of the figures were put into in the results. Figure 2 was re-structured for easier access to read. Please refer to line 314 to 321.
8	Figures (fig2-b,c,d,e) of high-throughput sequencing are not clear enough, please update them.	We have updated the figures with better resolution. Please refer to Fig. 2.
9	In the supplement document, the title should be placed below the picture, not above it.	The titles had been placed below each picture. Please refer to the supplementary file.
10	The author must first ask people from professional fields with excellent English writing ability to help rewrite the complete manuscript, and then ask professional English polishing service agencies to do the second round of proofreading. The current English writing is completely below the published standard.	We had linguistic service from a professional language polishing agent (Senior Editor Dr. Ivan Jakovlic, Bio-Transduction Lab). Please refer to the highlighted changes.
	Comments of reviewer 2	Reply
1	Methods and data for DUFAS were significantly lacking for both PK15 and fibroblasts. a. Was Neo selection applied?	Yes. Neo selection was carried out in DUFAS-mediated HDR targeting in all experiments, as is reflected in the targeting plasmid backbone that has the Neo selectable marker gene (supplementary Fig. 1).
2	b. What was the distribution of Neo + clones with one or both fluorescent markers?	We had statistic data regarding the distributuon of the cell clones after Neo selection. The avergae percentage for non-fluorescence cell clones was 27.4%, RFP-only 48.7%, both fluorecence 13.7%, and GFP-only 10.2%. A pie chart was added to the supplementary file to demonstrate this composition

		(Supplementary Fig. 3). We reasoned that RFP cassette is a mono-cistronic unit that independently expresses RFP, while GFP is expressed from 2A peptide downstream of Neo CDS, which is not comparable to that of RFP. Additionally, random integration is inevitable during selection, which might result in the transgene silencing due to positional effect. This is why non-fluorescent cell clones exist. In summary, DUFAS enables prompt differentiation of GFP-only cell clones out of the background.
3	c. What was the prevalence of perfect 5' and 3' integration in different classes of clones.	Homologous recombination is the error-free pathway to repair DSB. In all the sequenced clones of DUFAS-mediated targeting at the three sites, they displayed 100% accuracy of both 5' and 3' junctions, as shown in fig. 1c and supplementary fig. 2. In term of precision, DUFAS outplays indel-based gene editing.
4	d. For HDR positive clones in Sup Table 1, how were they validated? Were they confirmed at the 5', 3' junctions, or both?	Yes, HDR positive clones were validated by both 5' and 3' junction PCR as well as Sanger sequencing of the PCR products to prove the DNA sequences.
5	e. In Sup Fig 2 and Figure 1, are the "5 Clones" shown in the gel the same 5 clones from left to right for both 5' and 3' junctions?	Yes. The "5 clones" are loaded in the same order for both 5' and 3' junctions.
6	Several labs have reported high efficiency HDR using oligonucleotide templates that leave no remnants at the target site. When and why	There were indeed quite a few reports to make use of oligonucleotides to introduce

	would DUFAS be applied if that is the case? Some examples (Wang, 2016 NAR; Tan, 2013 PNAS).	highly efficient HDR in mammalian cells, but very few studies reported successful generation of gene edited livestock via this method. The reason is that oligonucleotide-mediated HDR fails to take advantage of antibiotics to purify targeted cell clones for somatic nuclear transfer (SCNT). It is recommended that when performing zygote microjection, oligonucleotide-mediated HDR could be used. When cell selection is required, DUFAS would be exploited.
7	Figure 2 is nearly impossible to read- I presume this is due to lower resolution of the reviewer PDF, but consider increasing font. I was not adequately able to assess 2d and 2e since they could not be read.	We have updated the figures with better resolution. Please refer to Fig. 2.
8	What factors lead to selection of SCD5? There were 4 other nodes more prominent on the network, so please clarify the rationale.	There were four other genes which are eminent as well in the network. We made literature investigation upon them to attempt to identify a direct link with fat deposition. It turned out that SCD5 was the only one that can be directly linked with fat metabolism. The other four genes didn't have such a predisposition. For example, Thy1 gene is a tumor suppressor associated with lymph node metastases. In this way these four genes were ruled out and SCD5 was targeted for subsequent study.
9	Under subheading 5, you made the following statement "Concomitantly, MEF2C downregulation reduced miR222 expression	This misinterpretation had been corrected in the text. Please refer to line 304.

	and this further lowered SCD5 expression, as verified by qPCR and immunoblotting (Fig. 5a, b).” Should this statement state that SCD5 expression was increased?	
--	---	--

Reviewers' Comments:

Reviewer #1:

Remarks to the Author:

I have accepted all the revisions you made in the manuscript. Your research is novel and integrates the current popular omics approach. In subsequent experiments, please pay more attention to the minimum sample size requirements in statistical methods. I think your manuscript meets the quality of publication.

Reviewer #2:

Remarks to the Author:

Recommendation is to accept.

Comments on the rebuttal:

All seem to be adequately addressed, however this reviewer disagrees that oligo-mediated HDR requires selection. Rates > 50% are routine, which is higher than reported using DUFAS in this study. Would one plan to put a pig in the food chain expressing Neo and GFP?

Minor:

Line 52: Clarify what is meant by "hybrid generations"

Line 82: "Extremely" seems like an adjective for the pre-site specific nuclease era.